# An Object-Based Markov Random Field Model with Anisotropic Penalty for Semantic Segmentation of High Spatial Resolution Remote Sensing Imagery

**Chen Zheng [1,2,3], Xinxin Pan [1], Xiaohui Chen [4,\*], Xiaohui Yang [1,3], Xin Xin [1] and Limin Su [1]**

[1] School of Mathematics and Statistics, Henan University, Kaifeng 475000, China; chen.zheng@unb.ca (C.Z.); 104753170656@vip.henu.edu.cn (X.P.); xhyang@henu.edu.cn (X.Y.); xinxin@henu.edu.cn (X.X.); 10100033@vip.henu.edu.cn (L.S.)
[2] Department of Geodesy and Geomatics Engineering, University of New Brunswick, Fredericton, NB E3B 5A3, Canada
[3] Data Analysis Technology Lab, Institute of Applied Mathematics, Henan University, Kaifeng 475000, China
[4] Technology Search Station, Henan University, Kaifeng 475000, China
\* Correspondence: 10100075@vip.henu.edu.cn; Tel.: +86-1503-904-6557

**Abstract:** The Markov random field model (MRF) has attracted a lot of attention in the field of remote sensing semantic segmentation. But, most MRF-based methods fail to capture the various interactions between different land classes by using the isotropic potential function. In order to solve such a problem, this paper proposed a new generalized probability inference with an anisotropic penalty for the object-based MRF model (OMRF-AP) that can distinguish the differences in the interactions between any two land classes. Specifically, an anisotropic penalty matrix was first developed to describe the relationships between different classes. Then, an expected value of the penalty information (EVPI) was developed in this inference criterion to integrate the anisotropic class-interaction information and the posteriori distribution information of the OMRF model. Finally, by iteratively updating the EVPI terms of different classes, segmentation results could be achieved when the iteration converged. Experiments of texture images and different remote sensing images demonstrated that our method could show a better performance than other state-of-the-art MRF-based methods, and a post-processing scheme of the OMRF-AP model was also discussed in the experiments.

**Keywords:** semantic segmentation; object-based Markov random field; anisotropic penalty matrix

---

## 1. Introduction

Image segmentation is one of the most important tasks in remote sensing image processing. Its purpose is to divide an image into some homogeneous regions. Specifically, semantic segmentation requires that each homogeneous region can represent one specific land object, such as urban, farmland, forest. Many approaches have been proposed for semantic segmentation in the last decades, including clustering methods [1–4], level set [5–7], deep learning [8–12], Markov random field model (MRF) [13–19], etc. During them, some early approaches, such as clustering methods, usually assume that pixels belonging to the same object should have similar characteristics, and pixels between different objects would show different appearances. This assumption works well for the low or medium spatial resolution remote sensing images but fails to apply to the high spatial resolution (HSR) remote sensing image. In the HSR images, pixels in the same object would have various appearances, and some pixels of different objects could have similar characteristics. Moreover, more sub-objects can be recognized in the HSR image, and different objects may have the same sub-objects. For instance, as shown in Figure 1a, the sub-object, trees, may belong to the urban or the forest.

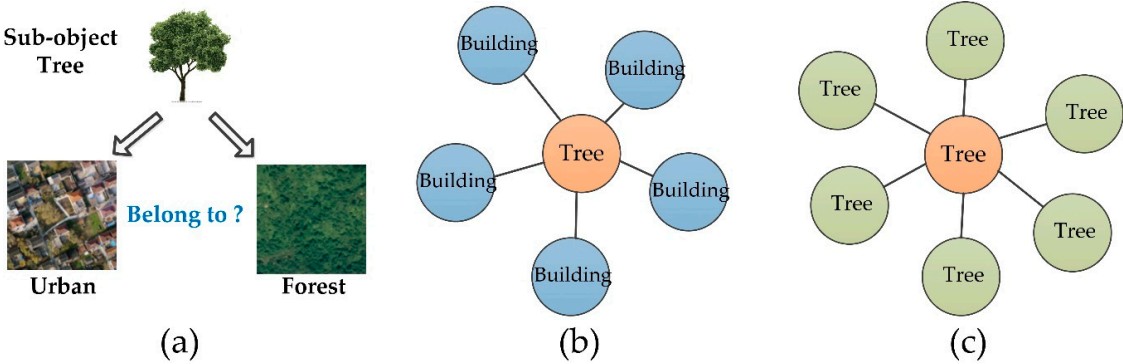

**Figure 1.** (**a**) An illustration of that one sub-object may belong to different objects, (**b**) the spatial relationship of a tree in the urban, (**c**) the spatial relationship of a tree in the forest.

Hence, the main challenges of semantic segmentation are how to effectively extract the semantic information and how to capture the homogeneity of sub-objects within one object and the heterogeneity of sub-objects between different objects in the HSR images.

To solve this problem, many approaches, such as level set [5–7], MRF [13–19], deep learning [8–12], introduce more constraints or prior information to improve the accuracy of semantic segmentation. These approaches can be roughly into three groups. One treats the semantic segmentation as the deterministic optimization problem, such as the level set. The second group is the deep learning-based methods that use the deep-layer neural network to get the segmentation by employing the convolution and related technical tricks. The last one formulates the semantic segmentation as a stochastic optimization problem. Our work belongs to the last one; especially, a new MRF-based method was proposed in this paper. For the MRF model, it has a very complete probability theoretical foundation based on the probability graph. With the Markov property, the MRF model can effectively integrate semantic features of a given image and its spatial neighborhood interactions and reduce the impact of intraclass variations. Hence, it is one of the most widely used methods for semantic segmentation. MRF is a probability graph model that uses a statistic way to model the spatial neighborhood relationships between pixels. There are two sub-models in the MRF model [13,19]. One is the feature field that can effectively extract and model different features with a likelihood function, which measures the probability of the occurrence for the features of one pixel. The other sub-model is the label field that uses the potential function and the Markov property to reduce the heterogeneity of pixels within the same object by modeling the spatial neighborhood interactions between the class labels of pixels. Under the framework of Bayesian statistic, both the information of the feature field and the label field are, respectively, treated as the observed information and the prior information, and then the posterior probability can be obtained by integrating them together, and the final segmentation can be provided according to the maximum a posteriori (MAP) criterion.

In the classic MRF model, the likelihood function of the feature field can only consider the pixel-based features, and the potential function of the label field can just model the spatial neighborhood interactions within a small spatial context, such as the 4-pixels or 8-pixels neighborhood. Hence, the classic MRF model has been extended to capture more complex structures within a larger neighborhood. In [20], Bouman and Liu used the quad tree to represent the original image with the multi-resolution pyramid structure, and a lattice pixel at one resolution would correspond to four points at the next finer resolution. Then, a Gaussian autoregressive model was used to describe the image, and the segmentation result could be achieved by sequentially maximizing the posteriori probability of each resolution. In [21], due to the nonredundant directional selectivity and discriminative nature of the wavelet representation, the wavelet transform was employed to provide the pyramid image decomposition, and the MRF model was defined over it. These multi-resolution MRF (MRMRF) methods can improve the classic MRF model by capturing image features and modeling the spatial context in a large neighborhood. But these methods are still pixel-based MRF methods,

which are difficult to effectively describe the macro spatial interactions of the HSR images. Hence, the object-based MRF (OMRF) methods [17,19,22–24] are studied in recent years. These OMRF methods use the over-segmented regions as the basic units of the MRF model, which can not only utilize the regional image features but also take macro spatial neighborhood interactions into account.

These optimized MRF-based methods, such as the MRMRF and OMRF models, indeed enhance the segmentation accuracy by extending the neighborhood, but their potential functions of the label field are isotropic. For instance, the Ising model, Potts model, multi-level logistic model [13] are commonly employed to define the isotropic potential function. These isotropic potential functions can only determine whether different sites in the probability graph belong to the same class, but cannot measure the relationships between different land classes. However, different objects of the HSR images are usually dependent, and their relationships are useful for semantic segmentation. To evaluate the relationships between different classes, Clausi et al. [25,26] used a weighted potential function by considering the edge information in the OMRF model. Ladicky et al. [27] proposed an associative hierarchical random field to consider the context between classes of different granularities. Wang et al. [28] proposed an anisotropic spatial energy function to describe the class co-occurrence dependency. These approaches optimize the potential function, but they are still the isotropic potential functions with respect to different weights.

To capture relationships between different classes, a new generalized probability inference with an anisotropic penalty for the OMRF model (OMRF-AP) was proposed in this paper. The OMRF-AP model first established a penalty matrix to reflect the anisotropic relationships of different land classes. Then, according to the anisotropic penalty matrix and the posterior probability of the OMRF model, an expected value of the penalty information (EVPI) of each class was developed for all the sites in the probability graph. Finally, the segmentation result was realized by minimizing the EVPI value. For the OMRF-AP, it not only considered the anisotropic class-information of the neighborhood with the penalty matrix but also integrated this information with the posterior probability in the inference criterion. It could break the limitation of the isotropic potential function. Furthermore, the solution of this model was achieved by minimizing the EVPI value instead of the maximum a posteriori (MAP) criterion. The proposed OMRF-AP has been introduced in Section 2. Section 3 discusses the parameter setting of the OMRF-AP and demonstrates experimental results. Section 4 is the discussion section, and the conclusion is given in Section 5.

## 2. The OMRF-AP Model for Image Segmentation

The proposed OMRF-AP model aimed to use the anisotropic penalty matrix to describe the differences between different classes for semantic segmentation. In this section, the MRF model has been briefly reviewed for segmentation, and then the OMRF-AP model has been introduced in detail.

### 2.1. MRF Model for Image Segmentation

Let $G = \{V, E\}$ denote the probability graph of the MRF model. Here, $V = \{V_s\}_{s \in S}$ is the set of vertexes, and $E = \{e_{st}\}_{s,t \in S}$ is the set of edges, where each $s$ denotes a site in the probability graph, and $S$ is the set of sites. If $V_s$ and $V_t$ are spatial adjacent, $e_{st} = 1$; otherwise, $e_{st} = 0$. Specifically, in the probability graph $G$, if each $s$ denotes a pixel, this graph is used for the classical pixel-based MRF model; and if each $s$ denotes an over-segmented region, then $G$ is used for the OMRF model. For instance, an example of the OMRF model is shown in Figure 2, where each circle denotes one over-segmented regional vertex, and the red line between two vertexes means that these two vertexes are adjacent. For the observed image $I = \{I_s | s \in S\}$, each $I_s$ is the observed data of $V_s$. The label field $X = \{X_s | s \in S\}$ is also defined on the $S$. Each random variable $X_s$ denotes the label class of $V_s$ and takes value from the set $\Lambda = \{1, 2, \ldots, k\}$, and $k$ is the number of the classes. If $x = \{x_s | s \in S\}$ is a realization of

$X$, its posteriori probability $P\{X = x|I\}$, under the condition of the observed image $I$, can be calculated based on the Bayesian formula. That is

$$P\{X = x|I\} = \frac{P\{I|X = x\}P\{X = x\}}{P\{I\}}$$

(1)

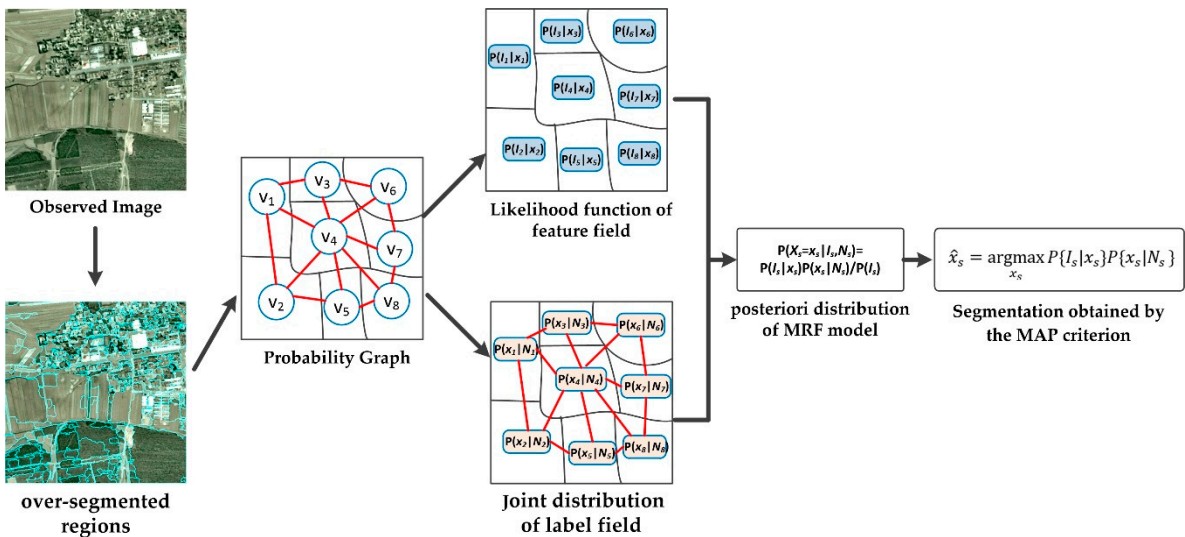

**Figure 2.** Flowchart of the OMRF (object-based Markov random field) model for image segmentation.

In the above equation, the likelihood function $P\{I|X = x\}$ can measure the probability of the observed image $I$ occurring in terms of the given realization $X = x$. It usually assumes this function obeys the naive Bayes assumption, that is

$$P\{I|X = x\} = \prod_{s \epsilon S} P\{I_s|X_s = x_s\}$$

(2)

The joint distribution $P\{X = x\}$ can capture the spatial neighborhood interactions between labels of different sites. It is of the Markov property, that is

$$P\{X_s|X_t, t \in V/V_s \} = P\{X_s|X_t, t \in N_s \}$$

(3)

Here, $N_s$ is the set of vertexes neighboring to $V_s$, i.e., vertex $V_t$ belongs to $N_s$ if $e_{st} = 1$. According to the Hammersley and Clifford theorem [13], the $P\{X = x\}$ is of the Gibbs distribution,

$$P\{X = x\} = \frac{1}{Z}e^{-U(x)}$$

(4)

and for each vertex $V_s$,

$$P\{X_s = x_s|X_t, t \in N_s\} = \frac{e^{-U(x_s)}}{\sum_{x'_s \epsilon \Lambda} e^{-U(x'_s)}}$$

(5)

Here, $U(x_s) = \sum_{c \epsilon C} V_c(x_s)$ is the energy function that equals to the sum of the potential functions $V_c(x_s)$ over all the possible cliques c.

According to the posteriori probability $P\{X = x|I\}$ of Equation (1), the segmentation of the given image $I$ can be realized by finding the optimal realization $\hat{x}$ with the MAP criterion, i.e.,

$$\hat{x} = \underset{x}{\mathrm{argmax}}P\{X = x|I\} = \underset{x}{\mathrm{argmax}}P\{I|X = x\}P\{X = x\}.$$

(6)

For image segmentation, pair-site cliques are usually used to calculate $U(x_s)$ in the $P\{X_s|X_t, t \in N_s\}$, i.e., $U(x_s) = \sum_{t \in N_s} V(x_s, x_t)$ and

$$V(x_s, x_t) = \begin{cases} -\beta & if\ x_s = x_t \\ \beta & if\ x_s \neq x_t \end{cases}, \tag{7}$$

where $\beta$ is the potential parameter. Hence, $P\{X = x\}$ can be represented as

$$P\{X = x\} = \prod_{s \in S} P\{X_s = x_s|X_t, t \in N_s\} = \prod_{s \in S} \frac{e^{-U(x_s)}}{\sum_{x'_s \in \Lambda} e^{-U(x'_s)}} \tag{8}$$

Hence, the optimal realization $\hat{x} = \{\hat{x}_s\}$ can be sequentially obtained by optimizing each $\hat{x}_s$, i.e.,

$$\hat{x}_s = \underset{x_s}{\operatorname{argmax}} P\{X_s = x_s|I_s, X_t, t \in N_s\} = \underset{x_s}{\operatorname{argmax}} P\{I_s|X_s = x_s\} P\{X_s = x_s|X_t, t \in N_s\} \tag{9}$$

### 2.2. Proposed OMRF-AP Model

In the classic OMRF model, the potential function $V(x_s, x_t)$ is designed to capture interactions between different classes of neighboring vertexes. But, the value of this isotropic function can only indicate whether two adjacent vertexes belong to the same class by taking $-\beta$ or $\beta$, as shown in Equation (7). However, a key issue of semantic segmentation is how to use spatial interactions to effectively describe the differences between different land objects. In order to capture more complex relationships between different classes, an anisotropic penalty matrix (APM) $A = \{A_{i,j}\}$, $1 \leq i,j \leq k$ was developed in this section, i.e.,

$$A = \{A_{i,j}\} = \begin{bmatrix} A_{1,1} & A_{1,2} & \cdots & A_{1,k} \\ A_{2,1} & A_{2,2} & \cdots & A_{2,k} \\ \vdots\ \vdots & & \ddots & \vdots \\ A_{k,1} & A_{k,2} & \cdots & A_{k,k} \end{bmatrix}. \tag{10}$$

In this matrix, for any class $i$ and $j$ ($1 \leq i,j \leq k$, $i \neq j$), their specific relationships can be denoted as $A_{i,j}$ and $A_{j,i}$. Here, we assumed the relationship between two classes is directed, i.e., $A_{i,j} \neq A_{j,i}$. Each $A_{i,j}$ denotes the penalty that the true land class is class $i$ and the segmented result is class $j$. Hence, if the $A_{i,j}$ takes a small value, that means class $i$ are closely related to class $j$. Otherwise, the large value of $A_{i,j}$ would indicate that the relationship between class $i$ and class $j$ is relatively alienated, and it is difficult to achieve the transformation between them.

According to the APM, if the true land class of one vertex $V_s$ is $X_s = i$ and the segmented class $\widetilde{X}_s$ is $j$, then the penalty of $V_s$ is $A(X_s = i, \widetilde{X}_s = j) = A_{i,j}$. Because the true class $X_s$ of each vertex $V_s$ is unknown during the segmentation progress, it is treated as a random variable, and the probability of its occurrence is represented by posteriori probability $P\{X_s = i|I_s, X_t, t \in N_s\}$. To measure the cost of labeling $V_s$ as class $j$, an expected value of the penalty information (EVPI) was designed in the OMRF-AP model, which was calculated as follows,

$$R_s(j) = E^{P\{X_s|I_s, X_t, t \in N_s\}} \left[A(X_s = i, \widetilde{X}_s = j)\right] = \sum_{i=1}^{k} A_{i,j} \cdot P\{X_s = i|I_s, X_t, t \in N_s\}. \tag{11}$$

In the EVPI term, $R_s(j)$ can reflect the average penalty that $V_s$ is marked as class $j$, which integrates both the anisotropic class-interaction information and the classic OMRF-based posteriori information. Based on this EVPI term $R_s(j)$, the appropriate class of $V_s$, i.e. $\hat{x}_s^{AP}$, should be the one that has the minimum penalty. It can be provided by finding the minimum EVPI values during all the $R_s(j)$, $j$=1,2, ... ,$k$. That is

$$\hat{x}_s^{AP} = \underset{j \in \{1,2,...,k\}}{\operatorname{argmin}} R_s(j). \tag{12}$$

On comparing this equation with Equation (9) of the MAP criterion, we could see that this inference criterion brought two main advantages. One was that the EVPI term $R_s(j)$ could capture not only the observed information and the spatial context with the posteriori probability but also the anisotropic interactions between various classes. The other was that the EVPI term could involve more posteriori information. In fact, the MAP criterion of the classic MRF only focuses on the class information $\hat{x}_s$ that maximizes the posteriori probability $P\{X_s = i|I_s, X_t, t \in N_s \}$, but the EVPI term would use all the class information of the posteriori probability $P\{X_s = i|I_s, X_t, t \in N_s \}$ for $V_s$ and integrate them with the corresponding anisotropic penalty. Hence, the inference criterion minimizing the EVPI provided a new feasible way to obtain the appropriate class of each vertex, and the segmentation result of the OMRF-AP model $\hat{x}^{AP}$ could be achieved by iteratively finding the optimal $R_s(j^*)$ until convergence, i.e., $\hat{x}^{AP} = \{\hat{x}_s^{AP}|\hat{x}_s^{AP} = \underset{j\epsilon\{1,2,...,k\}}{\mathrm{argmin}} R_s(j)\}$. The flowchart of the OMRF-AP is shown in Figure 3.

The properties of the anisotropic penalty matrix and the EVPI term were further discussed as follows.

(1)　The default value of $A\left(X_s = i, \widetilde{X}_s = j\right)$, i.e., $A_{i,j}$, is

$$A\left(X_s = i, \widetilde{X}_s = j\right) = \left\{ \begin{array}{ll} 0 & if \ i = j \\ 1 & if \ i \neq j \end{array} \right. . \tag{13}$$

In fact, we could prove that the optimal class obtained by the proposed inference criterion is the same as the class obtained by the MAP criterion if each $A\left(X_s = i, \widetilde{X}_s = j\right)$ in the matrix $A$ takes the default value. Namely, with the values of Equation (13), we have

$$
\begin{aligned}
\hat{x}_s^{AP} &= \underset{j\epsilon\{1,2,...,k\}}{\mathrm{argmin}} R_s(j) \\
&= \underset{j\epsilon\{1,2,...,k\}}{\mathrm{argmin}} \sum_{i=1}^{k} A_{i,j} \cdot P\{X_s = i|I_s, X_t, t \in N_s \} \\
&= \underset{j\epsilon\{1,2,...,k\}}{\mathrm{argmin}} \sum_{i \neq j} P\{X_s = i|I_s, X_t, t \in N_s \} \\
&= \underset{j\epsilon\{1,2,...,k\}}{\mathrm{argmin}} \left(1 - P\{X_s = j|I_s, X_t, t \in N_s \}\right) \\
&= \underset{j\epsilon\{1,2,...,k\}}{\mathrm{argmax}} P\{X_s = j|I_s, X_t, t \in N_s \} \\
&= \underset{x_s\epsilon\{1,2,...,k\}}{\mathrm{argmax}} P\{X_s = x_s|I_s, X_t, t \in N_s \} \\
&= \underset{x_s\epsilon\{1,2,...,k\}}{\mathrm{argmax}} P\{I_s|X_s = x_s\}P\{X_s = x_s|X_t, t \in N_s \} = \hat{x}_s
\end{aligned}
\tag{1}
$$

This means that the MAP criterion of the MRF model was a special case of the proposed inference criterion of the OMRF-AP method.

(2)　If $A\left(X_s = i, \widetilde{X}_s = j\right) > 1$, i.e., $A_{i,j} > 1$, it means that the current class of $V_s$ repels class $j$ compared with other classes. In general, this applies to the situation that class $i$ is easily misclassified as class $j$. It is suitable for correcting the misclassifications between two classes $i$ and $j$ if they have the same or similar subclasses. If $A\left(X_s = i, \widetilde{X}_s = j\right) < 1$, it means that the class of vertex $s$, except class $i$, prefers class $j$ compared with other classes.

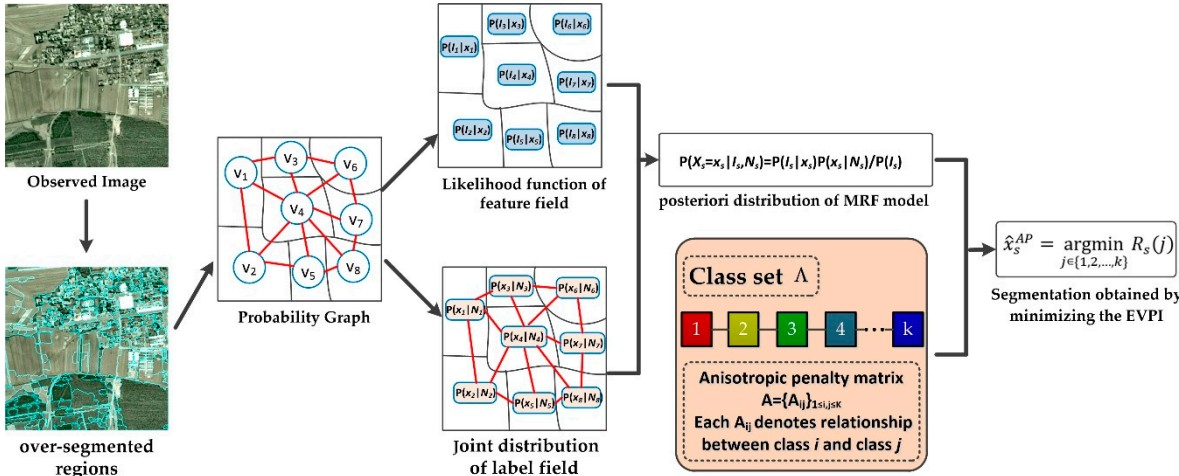

**Figure 3.** Flowchart of the proposed OMRF-AP (anisotropic penalty for the object-based MRF) model for image segmentation.

The algorithm of the OMRF-AP is given as follows (Algorithm 1).

---

**Algorithm 1.** OMRF-AP model

---

Input: the observed image *I*, the number of classes *k*, the anisotropic penalty matrix.
Output: the segmentation result

---

(1) Use mean shift provided by EDISON (http://www.wisdom.weizmann.ac.il/~{}bagon/matlab.html 2012) to get the over-segmented regions as the object vertexes and build the corresponding object probability graph $G = \{V, E\}$.

(2) Utilize the classical MRF to obtain a pixel-level segmentation $x^p$ and initialize the label field of the OMRF-AP as $x^0 = \{x_1^0, x_2^0, \dots, x_n^0 | x_s^0 = median(x_s^p), s = 1, 2, \dots, n\}$. Here, *n* is the number of vertexes, i.e., the number of over-segmented regions, $x_s^p$ is the set of pixel labels of $x^p$ in the object *s*, and *median* denotes the median operation.

(3) Set $t = 0$.

(4) Estimate parameters $\mu_h$ and $\Sigma_h$ of the likelihood function $P\{I_s | X_s = h\}$ based on Equation (15) and $x^t$, and update $P\{I_s | X_s = h\}$ of Equation (14) with these parameters.

(5) For each object *s*, calculate the clique potential $V(x_s, x_t)$ in Equation (7) based on $x^t$, and get the joint distribution $P\{X_s = x_s | X_t, t \in N_s \}$ based on Equation (8).

(6) Calculate each EVPI term $R_s(j)$ according to Equations (11) and (9), $j = 1, 2, \dots, k$, and update each $x_s^t$ as $x_s^{t+1}$ by finding the minimum EVPI terms, i.e.,

$$x_s^{t+1} = \operatorname*{argmin}_{j \epsilon \{1,2,\dots,k\}} R_s(j) = \operatorname*{argmin}_{j \epsilon \{1,2,\dots,k\}} \sum_{i=1}^{k} A_{i,j} \cdot P\{X_s = i | I_s, X_t, t \in N_s \}$$

$$= \operatorname*{argmin}_{j \epsilon \{1,2,\dots,k\}} \sum_{i=1}^{k} A_{i,j} \cdot P\{I_s | X_s = x_s\} P\{X_s = x_s | X_t, t \in N_s \}.$$

(7) Renew the label field $x^{t+1} = \{x_1^{t+1}, x_2^{t+1}, \dots, x_n^{t+1}\}$.

If $x^t = x^{t+1}$, stop and output the $x^{t+1}$ as the segmentation result;

else, $t = t + 1$, and go to step 4.

---

## 3. Experimental Results

The proposed OMRF-AP model provided a new way to introduce the anisotropic interactions between different classes into the classic MRF model for remote sensing image segmentation, which

could not only consider the spatial interactions but also the class interactions with the EVPI term. To experimentally evaluate this method, synthetic texture images and various HSR remote sensing images were tested in the following experiments. Two modules were discussed in this section. First, how to set the anisotropic penalty matrix (APM) and other parameters was discussed for the OMRF-AP model. Then, comparisons between the OMRF-AP and other MRF-based methods were demonstrated with different remote sensing images.

### 3.1. Parameter Settings of the OMRF-AP

In the proposed OMRF-AP model, a heuristic setting approach was designed to set the APM. Namely, there were $k(k-1)$ different $A_{i,j}$, $i \neq j$ in the APM, and their initial values were set as the default value of Equation (13). Then, the result was evaluated by the confusion matrix [29], and the term $A_{i,j}$ would be reset if the misclassification between class $i$ and $j$ had the maximum value during all the misclassifications in the confusion matrix. This process would continue until the accuracy rate of each class is higher than a given threshold. For instance, a SOPT5 image, located in Pingshuo, China, was tested with the threshold as 0.9 in Figure 4a. It included the agriculture field, vegetation, and urban area, which are denoted as class 1, 2, and 3, respectively. In this tested image, there were many vegetations in the urban area. Hence, these urban areas were wrongly recognized as the vegetation in the result with the default APM value, as shown in Figure 4d. According to the confusion matrix in Table 1, the accuracy rate of detecting an urban area was only 29.71% (<0.9), and the rate that urban areas misclassify as the vegetation was 51.21%, which had the maximum value during all the misclassifications.

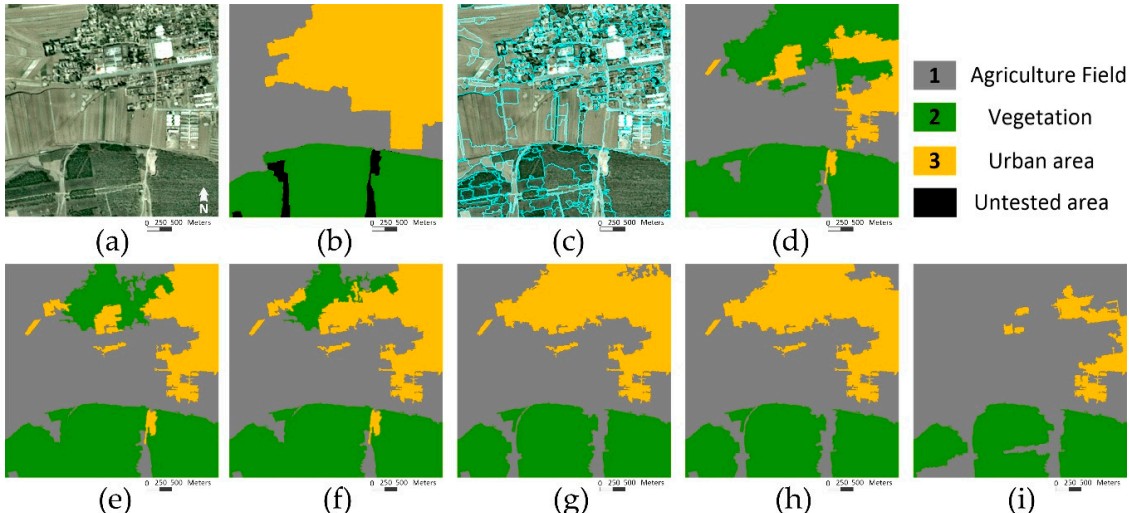

**Figure 4.** Example of setting the anisotropic penalty matrix (APM) value ($L_{3,2}$). (**a**) Original SPOT 5 image, (**b**) Visual interpretation result, (**c**) Over-segmented regions, (**d**) Result with the default APM (anisotropic penalty matrix) value, (**e**) Result with $A_{3,2} = 1.01$, (**f**) Result with $A_{3,2} = 1.015$, (**g**) Result with $A_{3,2} = 1.02$, (**h**) Result with $A_{3,2} = 1.03$, (**i**) Result with $A_{3,2} = 1.04$.

**Table 1.** Confusion matrix of segmentation result with the default APM (anisotropic penalty matrix) value.

|  | **Agriculture Fields** | **Vegetation** | **Urban** |
|---|---|---|---|
| Agriculture fields | 61303 (0.9430) | 1134 (0.0216) | 13310 (0.1908) |
| Vegetation | 3328 (0.0512) | 51334 (0.9780) | 35716 (0.5121) |
| Urban | 375 (0.0058) | 23 (0.0004) | 20723 (0.2971) |

Hence, according to the heuristic setting approach, $A_{3,2}$ in the APM, i.e., the class interaction between urban area and vegetation, was set with a large value to correct these misclassifications.

Namely, we set different $A_{3,2}$ values from 1 to 1.1 with step 0.01, their kappa coefficient and overall accuracy (OA) [29] are shown in Figure 5a, and some results with different $A_{3,2}$ values are shown in Figure 4e–i. As we could observe, when the $A_{3,2}$ value started to increase from 1, some misclassifications of the urban area were corrected, as shown in Figure 4e–g, and the kappa and OA indexes of the OMRF-AP method also had the significant increase during the first interval (1, 1.019), indicating that the introduction of the APM could enhance the tradition OMRF and improve the segmentation accuracy. Then, the OMRF-AP method showed an optimal stationary performance during (1.02, 1.032), as shown in Figure 4g,h. It means that the appropriate APM value was robust in this interval. Moreover, with a further increase of $A_{3,2}$, the accuracy of the OMRF-AP would gradually reduce.

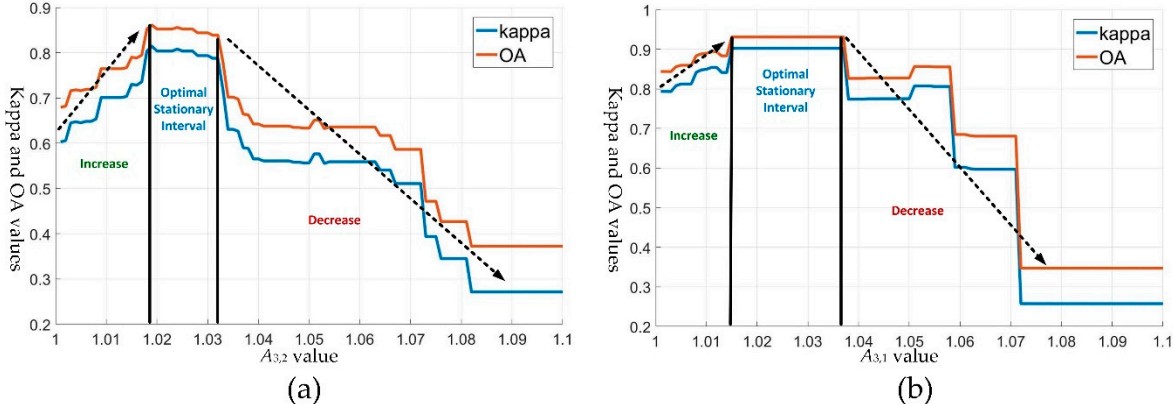

**Figure 5.** Kappa and OA (overall accuracy) values of different APM values. (**a**) Kappa and OA values of the OMRF-AP with different $A_{3,2}$ from 1 to 1.1 with step 0.001. (**b**) Kappa and OA values of the OMRF-AP with different $A_{3,1}$ from 1 to 1.1 with step 0.001 under the condition of $A_{3,2} = 1.03$.

Although misclassifications between vegetation and urban areas were corrected by setting $A_{3,2}$, the accuracy rate of detecting urban area was still less than 0.9, as shown in the confusion matrix of Table 2. It was due to that some urban areas were misclassified as the agriculture field, as shown in Figure 4h, which also had the maximum value during all the misclassifications in the current confusion matrix. Hence, according to the heuristic setting approach, $A_{3,1}$ in the APM, the class interaction between the urban area and agriculture field, was further adjusted under the condition of $A_{3,2} = 1.03$. Similar to $A_{3,2}$, different $A_{3,1}$ values from 1 to 1.1 were also tested with step 0.01. From Figure 5b, we could see that two quantitative indicators firstly increased in the interval (1,1.015), then had a robust optimal performance in the interval (1.015,1.036), and finally decreased in the interval (1.036,1.1). It showed the same trend as the curves of kappa and OA of $A_{3,2}$ and further improved the accuracy of the OMRF-AP model. Some results with different $A_{3,1}$ values are also demonstrated in Figure 6.

**Table 2.** Confusion matrix of segmentation result under the condition of $A_{3,2} = 1.03$.

|  | Agriculture Fields | Vegetation | Urban |
|---|---|---|---|
| Agriculture fields | 63058 (0.9700) | 3085 (0.0588) | 24186 (0.3468) |
| Vegetation | 985 (0.0152) | 49406 (0.9412) | 0 (0) |
| Urban | 963 (0.0148) | 0 (0) | 45563 (0.6532) |

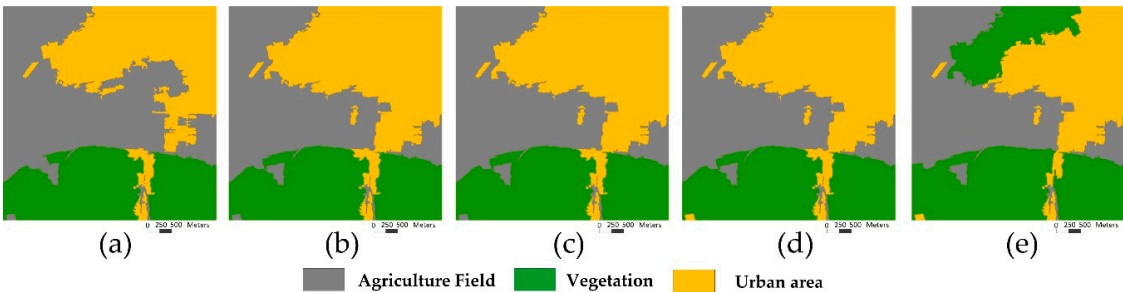

**Figure 6.** Example of setting the AMP value ($A_{3,1}$), under the condition of $A_{3,2} = 1.03$. (**a**) Result with $A_{3,1} = 1.01$, (**b**) Result with $A_{3,1} = 1.015$, (**c**) Result with $A_{3,1} = 1.02$, (**d**) Result with $A_{3,1} = 1.03$, (**e**) Result with $A_{3,1} = 1.04$.

With the setting $A_{3,2} = 1.03$ and $A_{3,1} = 1.02$, we could see that the accuracy of each class, i.e., the rate in the diagonal of the confusion matrix, was more than the threshold 0.9, as shown in Table 3. Hence, the heuristic setting approach could stop further exploring other $A_{i,j}$, $i \neq j$ in the APM, and the final APM was

$$A = \begin{bmatrix} 0 & 1 & 1 \\ 1 & 0 & 1 \\ 1.02 & 1.03 & 0 \end{bmatrix}.$$

**Table 3.** Confusion matrix of segmentation result under the condition of $A_{3,2} = 1.03$ and $A_{3,1} = 1.02$.

|  | **Agriculture Fields** | **Vegetation** | **Urban** |
|---|---|---|---|
| Agriculture fields | 60005 (0.9231) | 542 (0.0103) | 5724 (0.0821) |
| Vegetation | 994 (0.0153) | 50311 (0.9585) | 0 (0) |
| Urban | 4007 (0.0616) | 1638 (0.0312) | 64025 (0.9179) |

In addition to the APM value, there were parameters of the likelihood function $P\{I_s|X_s = h\}$ and the joint distribution $P\{X_s = x_s|X_t, t \in N_s\}$ in the OMRF-AP model that need to be set. Namely, the Gaussian mixture model was employed to define the likelihood function $P\{I_s|X_s = h\}$. Its probability distribution was denoted as

$$P\{I_s|X_s = h\} = (2\pi)^{-p/2}|\Sigma_h|^{-1/2} \exp\left\{-\frac{1}{2}\left[(I_s - \mu_h)^T \Sigma_h^{-1}(I_s - \mu_h)\right]\right\}. \tag{14}$$

Here, $\mu_h$ and $\Sigma_h$ are the mean and variance of the Gaussian distribution for the class $h$, $h \in \{1, 2, \ldots, k\}$, $I_s$ denotes the mean value of a pixel in region $s$, and $p$ is the dimension of the observed feature. The maximum likelihood estimation [30] could be used to evaluate these parameters. That is

$$\mu_h = \frac{\sum_{s \in S, X_s = h} \sum_{t \in I_s} I_t}{\sum_{s \in S, X_s = h} |I_s|}, \ \Sigma_h = \frac{\sum_{s \in S, X_s = h} \sum_{t \in I_s} (I_t - \mu_h)^T (I_t - \mu_h)}{\sum_{s \in S, X_s = h} |I_s|} \tag{15}$$

For the joint distribution $P\{X_s = x_s|X_t, t \in N_s\}$, the multilevel logistic model (MLL) [31], as shown in Equation (7), was used to define the potential function of the energy function. Similar to the discussion in literature [17,19], the potential parameter $\beta$ of the potential function was also quite robust to the OMRF-AP model by testing different values from 0 to 50 with step 0.5, as shown in Figure 7a. The optimal interval of $\beta$ was (0,20).

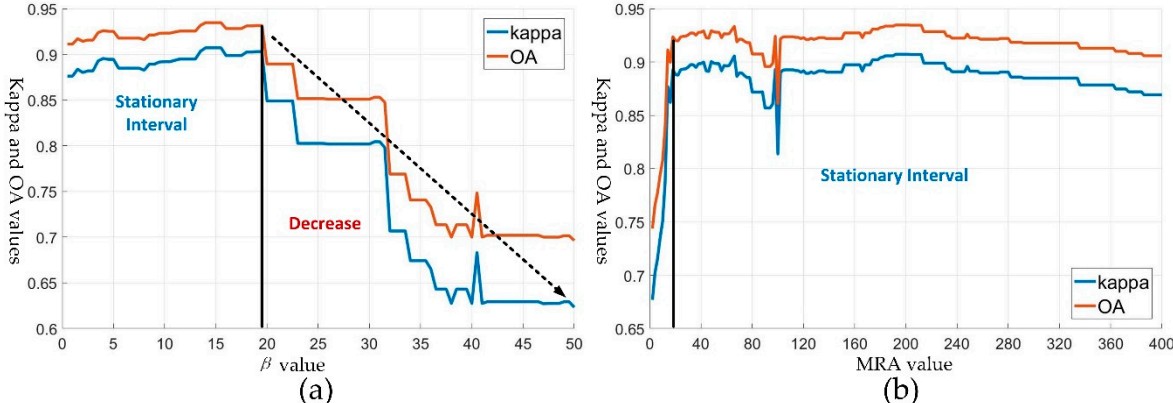

**Figure 7.** (**a**) Kappa and OA values of the OMRF-AP, with different $\beta$ values from 0 to 50, (**b**) Kappa and OA values of the OMRF-AP with different MRA (minimum region areas) values from 1 to 400.

Because the OMRF-AP was an object-based method, how to set the minimum region areas (MRA) of each object was another important issue. In the proposed method, the mean shift (MS) algorithm [32,33] was employed to get the initial over-segmented regions as the objects, and different MRA values were also used to test the robustness of MS by taking values from 1 to 400. As shown in Figure 7b, the OMRF-AP model was very robust to the degree of the MRA. Especially, it showed good performances when the MRA took value from 20 to 400, except for the slight fluctuations around 50.

In summary, although there were many parameters in the OMRF-AP model, some could be estimated according to the statistical method, such as parameters of Gaussian distribution; some could be set according to the developed heuristic setting approach, such as the APM value; and other parameters that need to be manually set were robust to the proposed method. They worked for the following experiments, as well.

### 3.2. Segmentation Experiments

In the following experiments, apart from the OMRF-AP model, five other state-of-the-art MRF-based methods were employed to compare the segmentation results, i.e., the iterated conditional mode (ICM) [14], the multiresolution MRF model (MRMRF) [21], the iterative region growing using semantics (IRGS) [25,26], the object-based MRF model (OMRF) [22], and a normalized Euclidean distance MRF model (NED-MRF) [28]. Here, the ICM was a classic pixel-level MRF model. The MRMRF used the wavelet transform to capture the spatial interactions at different spatial resolutions. The IRGS was an object-based MRF method that provided the result with the region growing scheme. The OMRF used an iterative probabilistic inference to find the segmentation result. The NED-MRF was a supervised method that used the support vector machine (SVM) to get the initial classification and developed an anisotropic spatial energy function to further improve the result. For the above object-based MRF methods, the mean shift algorithm was used to get the initial over-segmented regions. The mixed Gaussian distribution was used as the likelihood function for these methods.

#### 3.2.1. Segmentation of Texture Images

The remote sensing images contain rich texture information, so we first evaluated the effectiveness of the proposed method against texture. Two texture images were tested, in this subsection, that were generated by the Prague texture segmentation data generator [34]. These two images, as shown in Figures 8 and 9, were both sized 512 × 512 and contained six different texture patterns. For these texture patterns, each texture pattern showed different appearances, and the parts of various texture patterns had similar spectral values. This made segmentation challenging.

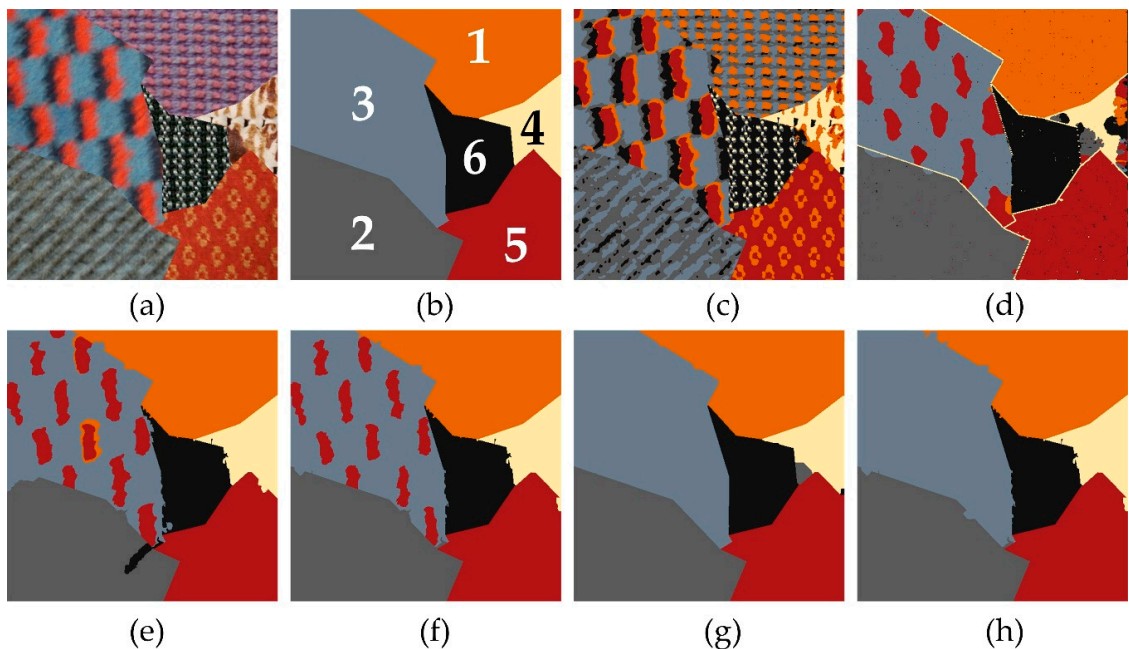

**Figure 8.** Segmentation results of six MRF-based methods for the first texture image. (**a**) Texture image (it is available as a Supplementary Material), (**b**) Ground truth with class number (each class number denotes one type of textures). (**c**) Result of ICM, (**d**) Result of MRMRF, (**e**) Result of IRGS, (**f**) Result of OMRF, (**g**) Result of NED-MRF, (**h**) Result of OMRF-AP. MRF: Markov random field, ICM: iterated conditional mode, MRMRF: multi-resolution MRF, IRGS: iterative region growing using semantics, NED-MRF: normalized Euclidean distance MRF model.

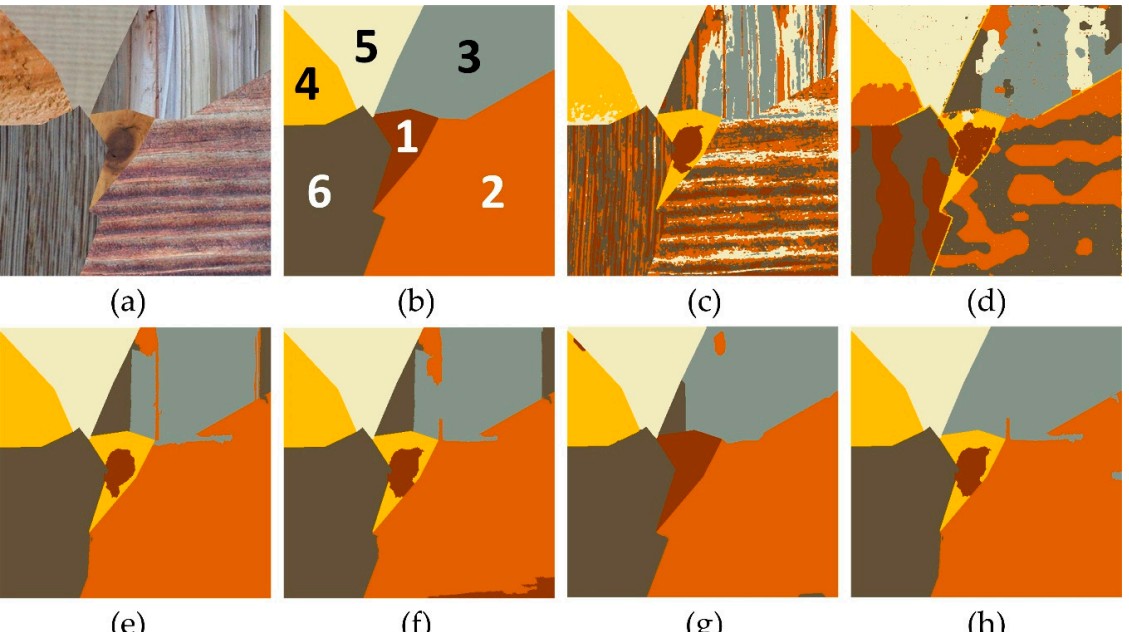

**Figure 9.** Segmentation results of six MRF-based methods for the second texture image. (**a**) Texture image (it is available as a Supplementary Material), (**b**) Ground truth with class number (each class number denotes one type of textures), (**c**) Result of ICM, (**d**) Result of MRMRF, (**e**) Result of IRGS, (**f**) Result of OMRF, (**g**) Result of NED-MRF, (**h**) Result of OMRF-AP.

The results of the six methods are demonstrated in Figures 8 and 9. For the ICM, the results could not effectively capture the texture patterns as it could only consider a small spatial neighborhood.

Hence, there were many misclassifications, as shown in Figures 8c and 9c, and the OA and kappa values were also the worst, as shown in Table 4. The MRMRF used the wavelet transform to extend the range of the neighborhood, outperforming the ICM. But it was a pixel-based method and also had many misclassifications, such as the result of Figures 8d and 9d. By using the over-segmented regions as the basic units, the IRGS and the OMRF, two object-based MRF methods, showed better performances than the pixel-based MRF methods. However, some pieces of misclassifications still existed, such as the texture at the left side (numbered 3) of Figure 8a and texture at the upper-right corner of Figure 9g. The NED-MRF further improved the segmentation by using the training data from the ground truth and the anisotropic spatial energy function. But, the definition of the anisotropic spatial energy function depended on the traditional potential function. Hence, it was actually a weighted isotropic function, and some misclassifications also existed, such as texture at the upper-right corner (numbered 3) of Figure 9a. The performance of the proposed OMRF-AP method, a method without training data, could reach or outperform the performance of the supervised NED-MRF, such as the texture at the upper-right corner of Figure 9h. That is to say, the APM and EVPI terms could provide more class-information to the OMRF model that showed a similar role in the information of the training data. APM could describe various interactions between different classes more accurately, and the new probability inference with the EVPI term could involve all the class information of the posteriori probability for each vertex. Thus, the OMRF-AP model could effectively distinguish texture patterns. The quantitative indicators of the OMRF-AP were also similar to the NED-MRF, as shown in Table 4. Furthermore, according to the heuristic setting approach, the APMs of these two texture experiments are given in the following Equation (16), where different texture patterns with class numbers are illustrated in the ground truth of each image, i.e., Figures 8b and 9b.

$$A_1(X_s, D_s) = \begin{bmatrix} 0 & 0.998 & 0.998 & 0.998 & 1 & 1 \\ 0.998 & 0 & 1 & 1 & 1 & 1 \\ 0.998 & 0.998 & 0 & 1 & 1 & 1 \\ 1 & 1 & 1 & 0 & 1 & 1 \\ 1.002 & 1 & 0.998 & 1.002 & 0 & 1 \\ 1 & 1.002 & 0.998 & 1.002 & 1 & 0 \end{bmatrix}, A_2(X_s, D_s) = \begin{bmatrix} 0 & 0.995 & 1 & 1 & 1 & 1 \\ 1.002 & 0 & 0.995 & 1 & 1 & 1 \\ 1 & 1 & 0 & 1 & 1 & 1 \\ 1 & 1 & 1 & 0 & 1 & 1 \\ 1 & 1 & 1 & 1 & 0 & 1 \\ 1 & 1 & 0.99 & 1 & 1 & 0 \end{bmatrix} \quad (16)$$

**Table 4.** Quantitative indicators of six comparison methods for six experiments.

| Indicator | Method | Figure 8 | Figure 9 | Figure 10 | Figure 11 | Figure 12 | Figure 13 |
|---|---|---|---|---|---|---|---|
| OA | ICM | 0.5113 | 0.5207 | 0.6635 | 0.7355 | 0.4344 | 0.6500 |
| | MRMRF | 0.8783 | 0.5287 | 0.8001 | 0.8754 | 0.4618 | 0.5322 |
| | IRGS | 0.9061 | 0.9166 | 0.7649 | 0.9191 | 0.8089 | 0.7505 |
| | OMRF | 0.9375 | 0.9096 | 0.7122 | 0.9285 | 0.8354 | 0.8100 |
| | NED-MRF | 0.9955 | 0.9823 | 0.9209 | 0.9381 | 0.8297 | 0.8564 |
| | OMRF-AP | 0.9941 | 0.9668 | 0.9311 | 0.9453 | 0.9149 | 0.9120 |
| Kappa | ICM | 0.4782 | 0.4866 | 0.5903 | 0.6572 | 0.3928 | 0.5518 |
| | MRMRF | 0.8548 | 0.4989 | 0.7515 | 0.8258 | 0.4193 | 0.4322 |
| | IRGS | 0.8877 | 0.8936 | 0.7027 | 0.8820 | 0.6970 | 0.6517 |
| | OMRF | 0.9240 | 0.8857 | 0.6424 | 0.8943 | 0.7283 | 0.7139 |
| | NED-MRF | 0.9943 | 0.9769 | 0.8889 | 0.9077 | 0.7265 | 0.7730 |
| | OMRF-AP | 0.9926 | 0.9566 | 0.9025 | 0.9170 | 0.8295 | 0.8486 |

OA: overall accuracy, ICM: iterated conditional mode, MRMRF: multiresolution MRF model, IRGS: iterative region growing using semantics, OMRF: object-based MRF model (OMRF), NED-MRF: a normalized Euclidean distance MRF model, OMRF-AP: a new generalized probability inference with an anisotropic penalty for the object-based MRF model.

### 3.2.2. Segmentation of Remote Sensing Images

To test the OMRF-AP model against real remote sensing images, experiments of four remote sensing images are illustrated in this section. The first one was a SPOT5 image, as shown in Figure 10a,

which sized 438 × 438 and was located in the Pingshuo area of China. It consisted of three objects, i.e., urban area, farmland, and vegetation. The results of the different methods are demonstrated in Figure 10. In this experiment, the APM of the OMRF-AP is given in Equation (17), and different objects with class numbers are illustrated in the visual interpretation result of Figure 10.

$$A_3(X_s, D_s) = \begin{bmatrix} 0 & 1 & 1 \\ 1 & 0 & 1 \\ 1.02 & 1.03 & 0 \end{bmatrix} \tag{17}$$

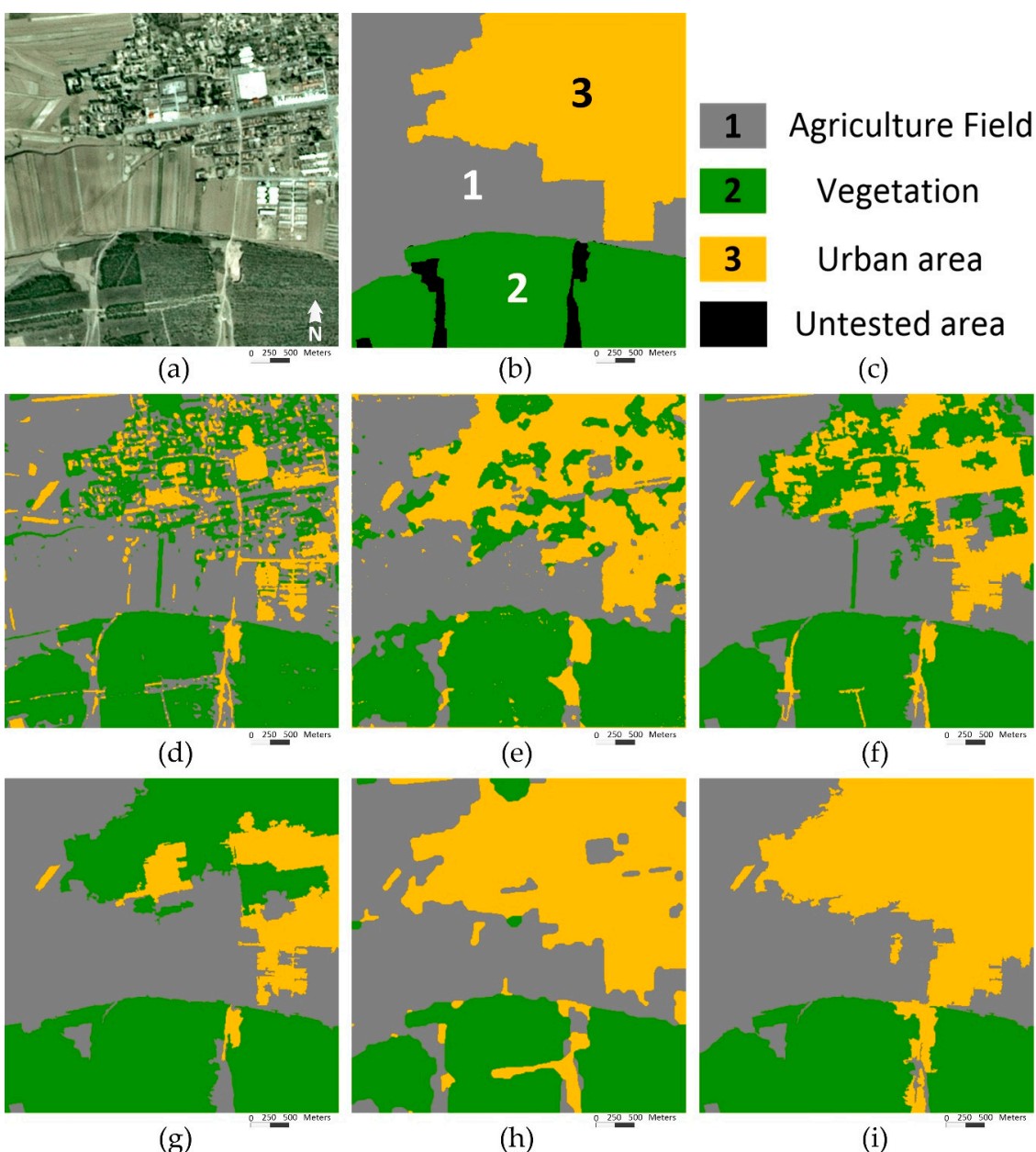

**Figure 10.** Segmentation results of the six MRF-based methods for the SPOT 5 remote sensing image. (**a**) SPOT 5 image (it is available as a Supplementary Material), (**b**) Visual interpretation result with class number, (**c**) Interpretation of each class label, (**d**) Result of ICM, (**e**) Result of MRMRF, (**f**) Result of IRGS, (**g**) Result of OMRF, (**h**) Result of NED-MRF, (**i**) Result of OMRF-AP.

As shown in Figure 10, the urban area of the SPOT5 image contained various sub-objects, such as roofs, roads, trees, showing different appearances. At the same time, there were trees in both the urban area and the vegetation. These factors led to many misclassifications of the urban area for the result of the ICM. By extending the spatial neighborhood, the MRMRF, IRGS, and OMRF performed much better. But, parts of the urban area were still wrongly labeled in their results. With the training data, the NED-MRF could roughly recognize the whole urban area. But it still failed to recognize some sub-objects with anomalous features, such as the building with the bright roof in the center of the urban area. By using the APM to specifically define the interaction between the urban area and other objects, the OMRF-AP could capture not only the spatial interactions but also the class interactions in the neighborhood. It made the OMRF-AP almost completely distinguish the urban area from the SPOT 5 image and showed a better performance than the performances of other comparison methods. The OMRF-AP also had the best performance of quantitative indicators in this experiment, as demonstrated in Table 4.

The second tested image, sized $2500 \times 2500$ with 0.2 m spatial resolution, was an aerial image of the Taizhou area, China, as shown in Figure 11a. This experiment would test the performance of the OMRF-AP model in the very high spatial resolution (VHSR) image. This VHSR image contained water, urban area, and the farmland, whose class numbers are illustrated in Figure 11b. The value of the APM is shown in the following equation for the OMRF-AP, i.e.,

$$A_4(X_s, D_s) = \begin{bmatrix} 0 & 1.002 & 1.002 \\ 1 & 0 & 0.98 \\ 1 & 1 & 0 \end{bmatrix}. \tag{18}$$

As shown in Figure 11, the aerial image not only had three different objects but also was large and had a spatial resolution. Hence, there were more detailed information and the long-term interactions between labels in this image. The ICM and the MRMRF had many small pieces of misclassifications as their spatial neighborhood was relatively small. The IRGS and the OMRF could improve the results by considering the larger neighborhood, but still had misclassifications, such as the area within the red ellipse in the lower-left corner. For the NED-MRF, although the training data helped this method to recognize the sub-object in the lower-left corner, its spatial energy function was constructed to regular pixels. Hence, when the tested image had large size and high spatial resolution, there were some small pieces of misclassifications in the result, as demonstrated in Figure 11h. The proposed OMRF-AP model could capture the long-term spatial interactions with object-based units and characterize the differences between various classes with APM, and then the new inference criterion with the EVPI term integrated all the information. Hence, it provided a better result with several large homogeneous regions, as shown in Figure 11i, indicating that the OMRF-AP could have a good performance on the large-size image, as well.

To further test the performance of the OMRF-AP against the VHSR image, another aerial image with 0.1 m spatial resolution was tested, as shown in Figure 12. It sized $2500 \times 2500$ and located in Taizhou, as well. There were water, urban areas, and farmland in this image. With the increasement of the spatial resolution, more detailed and diverse characteristics were demonstrated for each land class. Due to this, the results of the ICM and MRMRF had many misclassifications. By considering the over-segmented regions as the basic unit, the IRGS and OMRF extended the spatial neighborhood and provided more consistent results. But some misclassifications still existed, such as the buildings with red roofs in the urban area. With the training data, the NED-MRF could roughly distinguish three land classes. However, its pixel-based spatial energy function could not describe large spatial interactions effectively, which led to the wrong labeling of some shadow areas of the urban areas as the water. The OMRF-AP provided better performance than other methods by considering both the spatial interaction and class interaction in the object-based neighborhood. Namely, the long-term object-based spatial interactions helped the proposed method to get the rough segmentation, and the class interaction with APM $A_5(X_s, D_s)$ could further correct some misclassifications by considering the

relationship between specific land classes. The OMRF-AP also had the best quantitative indicators, as shown in Table 4.

$$A_5(X_s, D_s) = \begin{bmatrix} 0 & 1.01 & 1 \\ 1 & 0 & 1.02 \\ 1.02 & 0.98 & 0 \end{bmatrix}. \tag{19}$$

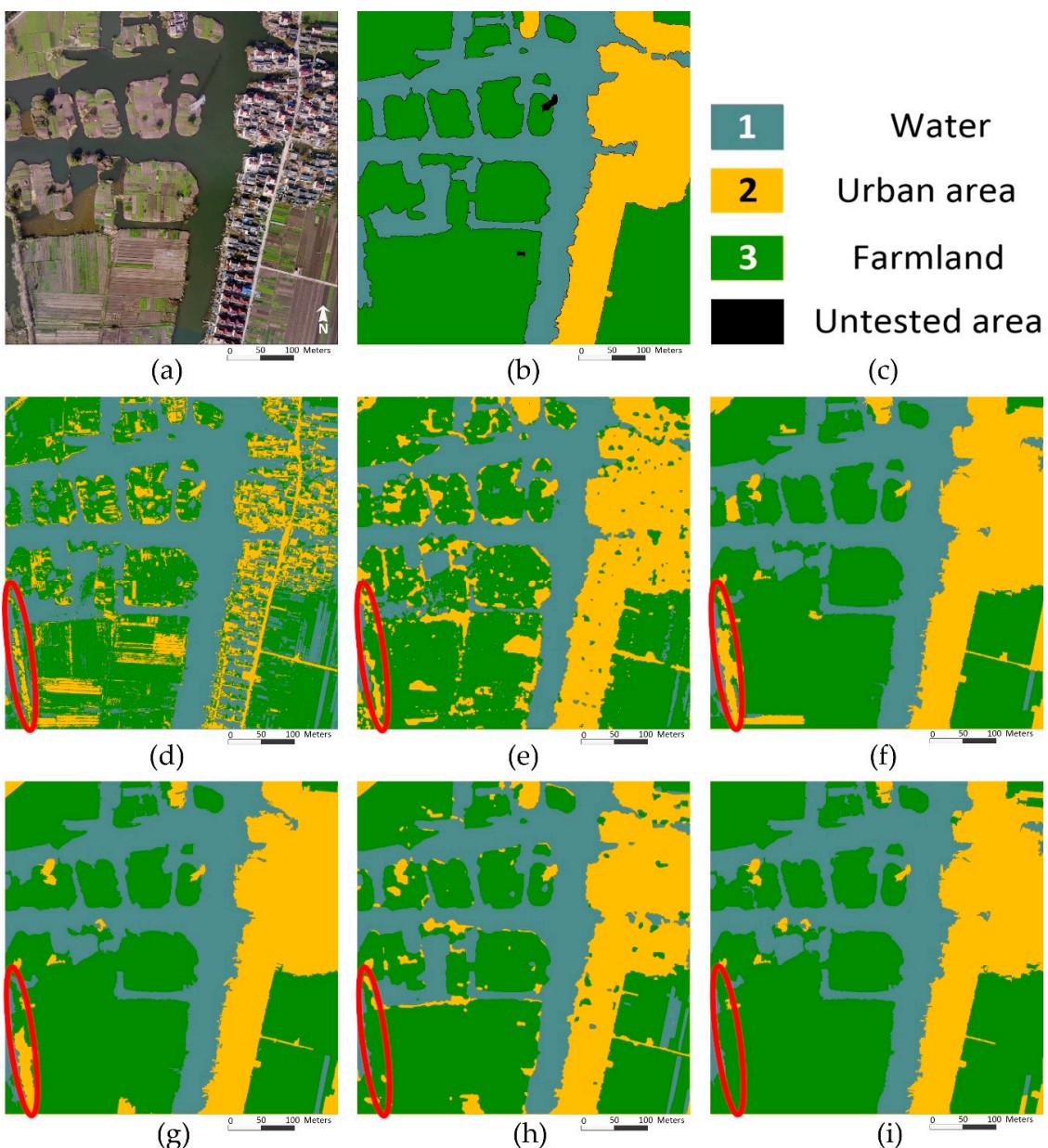

**Figure 11.** Segmentation results of the six MRF-based methods for the VHSR (very high spatial resolution) aerial remote sensing image. (**a**) Observed aerial image (it is available as a Supplementary Material), (**b**) Visual interpretation result with class number, (**c**) Interpretation of each class label, (**d**) Result of ICM, (**e**) Result of MRMRF, (**f**) Result of IRGS, (**g**) Result of OMRF, (**h**) Result of NED-MRF, (**i**) Result of OMRF-AP.

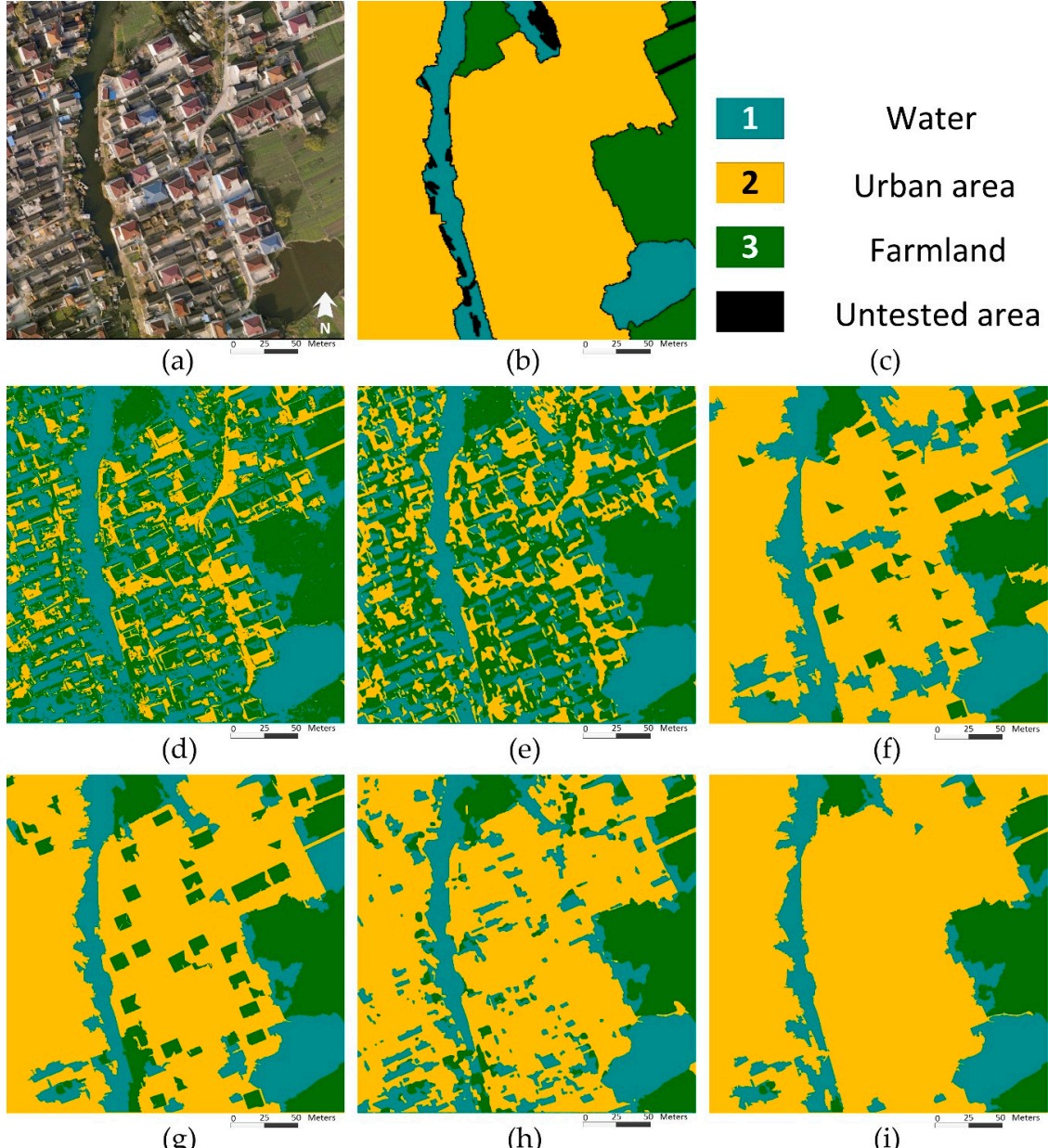

**Figure 12.** Segmentation results of the six MRF-based methods for the second VHSR aerial remote sensing image. (**a**) Observed aerial image (it is available as a Supplementary Material), (**b**) Visual interpretation result with class number, (**c**) Interpretation of each class label, (**d**) Result of ICM, (**e**) Result of MRMRF, (**f**) Result of IRGS, (**g**) Result of OMRF, (**h**) Result of NED-MRF, (**i**) Result of OMRF-AP.

In order to examine the performance of the proposed OMRF-AP in a remote sensing dataset, a Gaofen-2 remote sensing image was employed from the Gaofen image dataset (GID) [35]. The tested image was sized $1500 \times 1500$ with 3.2 m spatial resolution, located in Xining city, China, as demonstrated in Figure 13a. According to the ground truth, the tested land classes were urban area and vegetations, and the hills and other areas were untested, as shown in Figure 13b,c. The value of the APM was

$$A_6(X_s, D_s) = \begin{bmatrix} 0 & 1.02 & 1.01 \\ 1 & 0 & 1 \\ 1 & 1 & 0 \end{bmatrix}. \tag{20}$$

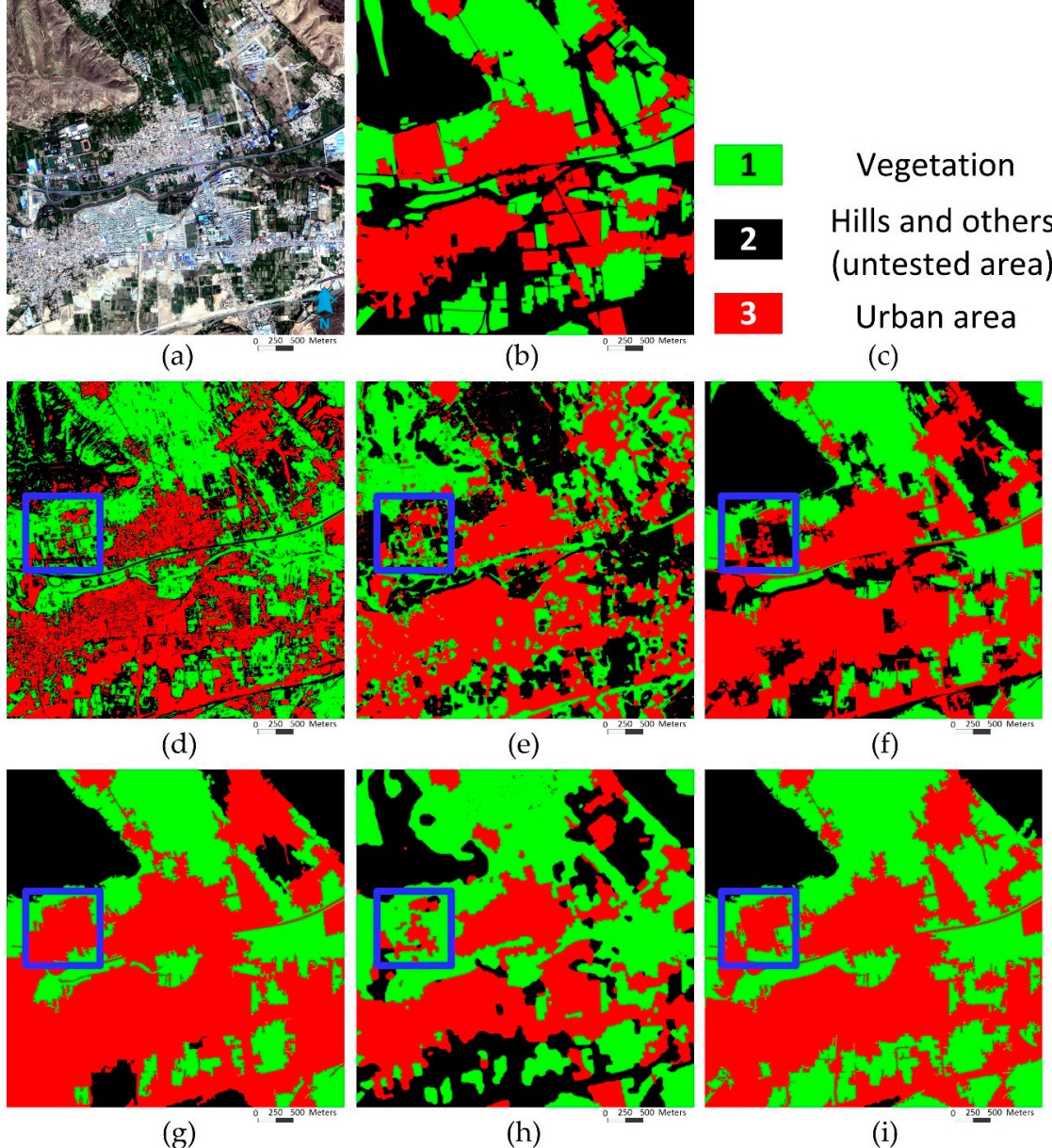

**Figure 13.** Segmentation results of the six MRF-based methods for the Gaofen-2 remote sensing image from GID (Gaofen image dataset). (**a**) Gaofen-2 image (it is available as a Supplementary Material), (**b**) Ground Truth, (**c**) Interpretation of each class label, (**d**) Result of ICM, (**e**) Result of MRMRF, (**f**) Result of IRGS, (**g**) Result of OMRF, (**h**) Result of NED-MRF, (**i**) Result of OMRF-AP.

As the differences between urban areas and vegetations are quite vague and difficult to distinguish in some places, it makes the semantic segmentation more challenging. For instance, the results of the ICM, MRMRF, IRGS, and NED-MRF failed to recognize the district of urban areas marked with a blue square as their appearance was similar to the vegetation, as shown in Figure 13d–f,h. The OMRF wrongly labeled some vegetations as urban areas and had an over-recognition about the urban area at the marked region and other regions. By considering the class interaction, the OMRF-AP showed a better performance than other comparison methods. The quantitative indicators in Table 4 also validated the superiority of the proposed method.

### 3.3. Post-Processing with Pixel-Based MRF

As mentioned in the literature [28,36–38], the result of the object-based method usually had the rough and blurred boundaries between different objects. The proposed OMRF-AP was an object-based method, and had a similar phenomenon in the results, as shown in Figure 14a–c,g–i. To correct these boundaries, a feasible way is to introduce the pixel-based method with detailed information as the post-processing [28,36,37]. In this section, the classic pixel-based MRF method, ICM, was employed as the post-processing module for the results of the OMRF-AP in the above experiments. Results of OMRF-AP with post-processing (OMRF-APP) are demonstrated in Figure 14 d–f,j–l.

As we could observe, the post-processing would not change the main part of previous results and just focus on the correction of boundaries. In fact, by comparing boundaries circled in red between OMRF-AP and OMRF-APP in Figure 14, one could see that some rough boundaries were smoothed by the post-processing, especially boundaries in Figure 14a,i. All the quantitative indicators could be slightly improved, as well, as shown in Table 5. Hence, the post-processing could indeed optimize the result of the OMRF-AP method. Please note that the post-processing was not a necessary step of the OMRF-AP, and this optional step was used according to the need of the application.

**Table 5.** Quantitative indicators of three object-based methods and their post-processing results for six previous experiments.

| Indicator | Method | Figure 8 | Figure 9 | Figure 10 | Figure 11 | Figure 12 | Figure 13 |
|---|---|---|---|---|---|---|---|
| OA | IRGS | 0.9061 | 0.9166 | 0.7649 | 0.9191 | 0.8089 | 0.7505 |
| | IRGS-P | 0.9153 | 0.9179 | 0.7669 | 0.9206 | 0.8097 | 0.7666 |
| | OMRF | 0.9375 | 0.9096 | 0.7122 | 0.9285 | 0.8354 | 0.8100 |
| | OMRF-P | 0.9430 | 0.9111 | 0.7165 | 0.9302 | 0.8362 | 0.8228 |
| | OMRF-AP | 0.9941 | 0.9668 | 0.9311 | 0.9453 | 0.9149 | 0.9120 |
| | OMRF-APP | 0.9998 | 0.9690 | 0.9328 | 0.9494 | 0.9165 | 0.9264 |
| Kappa | IRGS | 0.8877 | 0.8936 | 0.7027 | 0.8820 | 0.6970 | 0.6517 |
| | IRGS-P | 0.8983 | 0.8951 | 0.7035 | 0.8840 | 0.6980 | 0.6688 |
| | OMRF | 0.9240 | 0.8857 | 0.6424 | 0.8943 | 0.7283 | 0.7139 |
| | OMRF-P | 0.9305 | 0.8875 | 0.6478 | 0.8966 | 0.7292 | 0.7294 |
| | OMRF-AP | 0.9926 | 0.9566 | 0.9025 | 0.9170 | 0.8295 | 0.8486 |
| | OMRF-APP | 0.9998 | 0.9594 | 0.9048 | 0.9229 | 0.8325 | 0.8706 |

IRGS-P: results of IRGS with post-processing, OMRF-P: results of OMRF with post-processing, OMRF-APP: results of OMRF-AP with post-processing.

Because this post-processing module was developed for object-based methods, it was further tested for previous experimental results of the other two object-based comparison methods, i.e., the IRGS and OMRF. Quantitative indicators of their post-processing results are also illustrated in Table 5, which were denoted as the IRGS-P and OMRF-P, respectively. Similar to the OMRF-APP, the post-processing module could also slightly improve the original segmentation results of these two methods. To evaluate the efficiency of the post-processing module, the computational times of both original object-based methods and their post-processing modules are illustrated in Table 6, where the computational time of each original object-based method was the value before the plus sign, and the computational time of each post-processing module was the value after the plus sign. For instance, in Table 6, '12.36 + 0.93' of IRGS for Figure 8 means that the computational time of original IRGS was 12.36 seconds, and the post-processing time was 0.93 seconds. In general, the post-processing module would only increase a small amount of computational time.

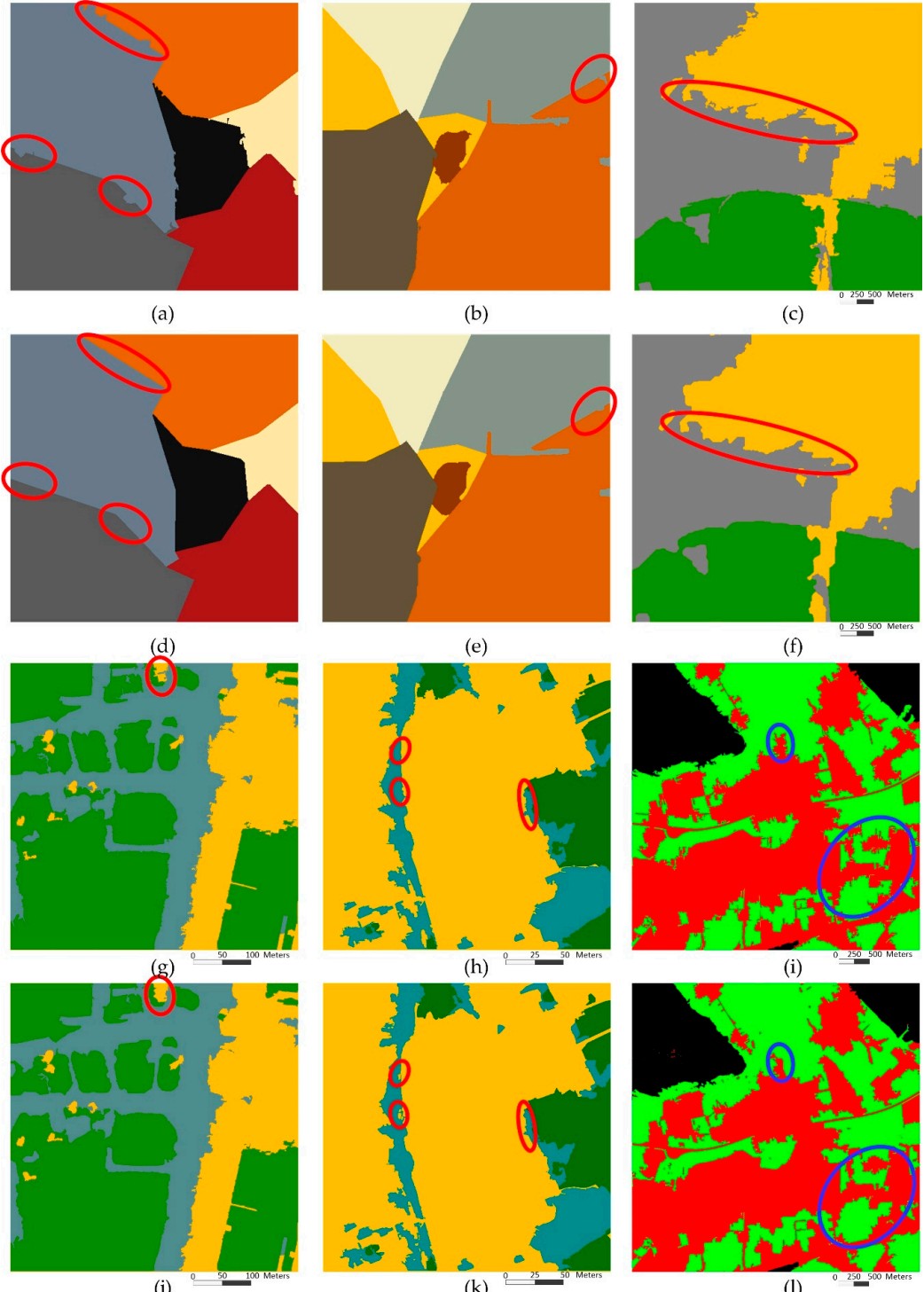

**Figure 14.** Segmentation results of OMRF-AP and OMRF-APP (OMRF-AP with post-processing) for four previous experiments. (**a**) and (**d**) Segmentation results of OMRF-AP and OMRF-APP for the first texture image (explanation of each texture please refers to Figure 8). (**b**) and (**e**) Segmentation results of OMRF-AP and OMRF-APP for the second texture image (explanation of each texture please refers to Figure 9). (**c**) and (**f**) Segmentation results of OMRF-AP and OMRF-APP for the SPOT 5 image (explanation of each class please refers to Figure 10). (**g**) and (**j**) Segmentation results of OMRF-AP and OMRF-APP for the first aerial image (explanation of each class please refers to Figure 11). (**h**) and (**k**) Segmentation results of OMRF-AP and OMRF-APP for the second aerial image (explanation of each class please refers to Figure 12). (**i**) and (**l**) Segmentation results of OMRF-AP and OMRF-APP for the Gaofen-2 image (explanation of each class please refers to Figure 13).

**Table 6.** The computational time of six methods and post-processing time of object-based methods (in seconds).

| Method | Figure 8 | Figure 9 | Figure 10 | Figure 11 | Figure 12 | Figure 13 |
|--------|----------|----------|-----------|-----------|-----------|-----------|
| ICM | 1.19 | 2.09 | 0.62 | 29.11 | 25.11 | 10.78 |
| MRMRF | 28.24 | 28.39 | 13.06 | 407.41 | 406.61 | 134.74 |
| IRGS | 12.36 + 0.93 | 19.45 + 0.92 | 8.45 + 0.47 | 375.90 + 14.77 | 438.86 + 14.85 | 140.52 + 5.16 |
| OMRF | 17.66 + 0.93 | 7.29 + 0.95 | 5.60 + 0.46 | 230.81 + 14.68 | 416.96 + 14.90 | 121.32 + 5.16 |
| NED-MRF | 27.02 | 33.01 | 12.21 | 363.40 | 552.82 | 157.60 |
| OMRF-AP | 20.45 + 0.93 | 7.34 + 0.93 | 7.58 + 0.46 | 232.70 + 14.74 | 376.86 + 14.85 | 80.06 + 5.23 |

*3.4. Computational Time*

The computational complexity of the OMRF-AP model was O($knt$), where $k$ is the number of the classes, $n$ is a number of the vertexes, and $t$ is the number of iterations. That is to say, the computational complexity of the OMRF-AP was the same as the OMRF model. In fact, for experiments in this paper, the OMRF-AP and the OMRF usually had the same number of the iterations. But, for each iteration, because the OMRF-AP took the anisotropic penalty matrix and the EVPI term into account, its computing time would be slightly higher than the time of the OMRF. The increased time would not exceed 0.15 seconds. The total time difference between the OMRF-AP and the OMRF was less than 3 seconds, as shown in Table 6.

In this paper, all the experiments were performed on the Windows 10 personal computer with an Intel i5-7300 CPU using 16-GB memory. The computing time of each method is illustrated in Table 6. As we could observe, the ICM ran faster than others due to it only considering a small spatial neighborhood. The MRMRF and the NED-MRF took a lot of time as one needed to spend time on the wavelet transform and the other to train the SVM with the sampling data. The time of the IRGS depended on the region's growing scheme. Sometimes it would be fast, but sometimes it would not. The OMRF and the OMRF-AP had a similar computing time. Hence, the speed of the proposed method was acceptable.

## 4. Discussion

In the above section, experimental results verified the superiority of the OMRF-AP. The OMRF-AP showed the following advantages. First, it was an object-based method that could model the long-term spatial interactions, which was particularly effective when dealing with the HSR image. The pixel-based methods, such as ICM, MRMRF, and NED-MRF, would suffer the inefficient use of spatial information, especially for the HSR remote sensing images. Second, the APM could describe the specific interactions between land classes, helping the proposed method capture information between land classes more accurately. Hence, when other object-based MRF methods, such as IRGS and OMRF, were trapped into the local optimum with the isotropic potential function, the OMRF-AP could break the limitation and obtain a more consistent result. At last, the new generative probability inference could effectively integrate spatial interactions and class interactions by minimizing the EVPI term.

Please note that, in the texture experiments, the NED-MRF showed a similar or even slightly better performance than the OMRF-AP. The patterns of texture images were relatively regular, which could be effectively learned by the training data of the NED-MRF. But, when it was used to the real remote sensing images, the NED-MRF showed poor performance than the proposed OMRF-AP as the texture information of remote sensing images was very irregular, especially for the HSR or VHSR images. Compared with the NED-MRF, the OMRF-AP showed good and stable performances in both texture experiments and remote sensing experiments. This was because the proposed method described the relationship between classes with APM, which worked well under no matter regular or irregular texture patterns.

The main contribution of the OMRF-AP was to verify that the anisotropic class information could improve the segmentation accuracy of current MRF-based methods with isotropic potential function,

which was validated by the above experiments. Although the OMRF-AP model could show good performances, its performance depended on the priori information about the number of classes and the APM, as demonstrated in the algorithm. That is to say, the proposed method was a semi-supervised method that needed to appropriately set the number of land classes and the APM. For the number of classes, it was usually the priori information for most MRF-based methods, and it could be properly set according to the purpose of segmentation. The APM needed to be set manually. Namely, one could get the optimal APM value based on the heuristic setting approach if one would have the ground truth, as illustrated in the above experiments. But we usually don't have the ground truth when we deal with the unknown data. Under these circumstances, we could experimentally initialize the APM value as an existed APM value whose image would be similar to the unknown data. For instance, a QuickBird remote sensing image was tested without ground and training data, as shown in Figure 15a. There were urban areas, farmland, and vegetation in this unknown data, which was similar to the land classes of SPOT5 image of Figure 10. Hence, the same APM value, $A_3(X_s, D_s)$, was employed for this image, and an appropriate result could be achieved with this APM, as demonstrated in Figure 15b.

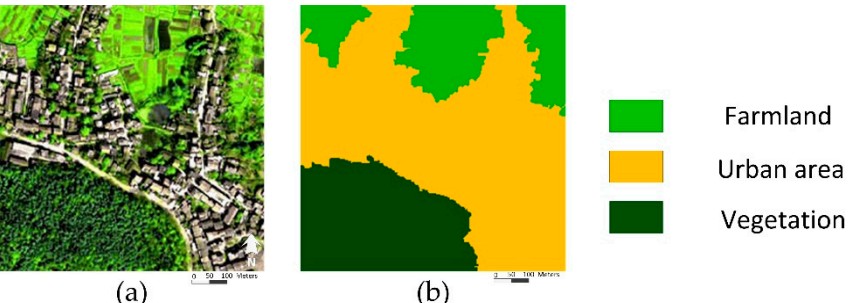

(a)                   (b)

**Figure 15.** Segmentation results of OMRF-AP of a QuickBird remote sensing image with the APM $A_3(X_s, D_s)$. (**a**) QuickBird image (it is available as a Supplementary Material), (**b**) Result of OMRF-AP with the APM $A_3(X_s, D_s)$.

In fact, the APM was used to reflect the relationship between certain land classes, and the geoscience knowledge could guide the setting of APM. When the geoscience relationships were explored, the number of class and the APM could be manually set for the unknown data without ground, especially when it was similar to known tested data. Moreover, as mentioned in Section 3.1, the optimal setting of the APM value was a relatively large interval instead of a single value. Hence, the APM value was quite robust, and it usually took value in the interval [0.98,1.03]. When we would deal with some new unknown data, we could also initialize its APM value in this interval. Furthermore, suitable APMs were also not unique. For instance, many urban areas were wrongly labeled as vegetation in the OMRF result without the APM of Figure 10; one could set $A_{3,2}$ in the APM with a large value, 1.03, to prevent the labeling of urban area as the vegetation, and set $A_{2,3}$ in the APM with a small value, 0.98, to encourage the correction of some vegetation to the urban area at the same time. Its APM value $(X_s, D_s)$ could be changed to

$$A\prime_3(X_s, D_s) = \begin{bmatrix} 0 & 1 & 1 \\ 1 & 0 & 0.98 \\ 1 & 1.03 & 0 \end{bmatrix}.$$

The OMRF-AP could also get an appropriate result with $A\prime_3(X_s, D_s)$, as shown in Figure 16b. For this result, its OA value was 0.9282, and the kappa value was 0.8987, which was close to the quantitative indicators of the result with $A_3(X_s, D_s)$. Hence, although the APM had many parameters, in addition to the heuristic setting approach, as mentioned above, one could initialize the APM more flexible. Compared with other semantic segmentation methods, such as deep learning, the OMRF-AP

had a complete theoretical foundation, and its APM had a clear and reasonable semantic interpretation. The proposed method was also very efficient as it did not need training processing.

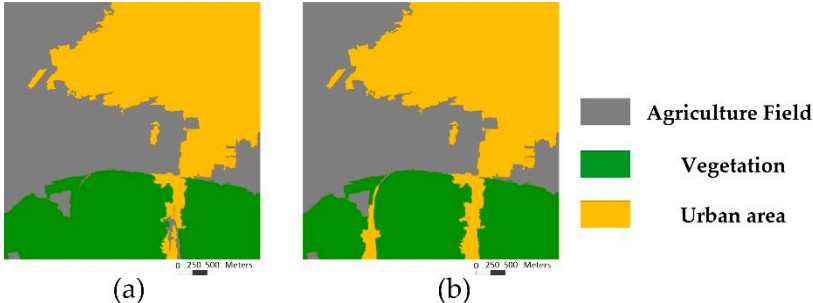

**Figure 16.** Segmentation results of OMRF-AP of the SPOT 5 remote sensing image with different APM. (**a**) Result of OMRF-AP with the original APM $A_3(X_s, D_s)$, (**b**) Result of OMRF-AP with a new APM $A'_3(X_s, D_s)$.

## 5. Conclusions

In this paper, an object-based Markov random field model with anisotropic penalty matrix was proposed for the semantic segmentation of the remote sensing images. The main contributions of the OMRF-AP model lied in two aspects. First, the APM was introduced into the MRF model to capture the various relationships between different classes. Second, the EVPI term, $R_s(j)$, replaced the posteriori probability to find the solution of the OMRF model. By considering both the spatial interactions and the class interactions in the neighborhood, the OMRF-AP model provided an optimized object-based MRF-based method. The effectiveness of the OMRF-AP was validated by experiments obtained from different texture images and remote sensing images.

Compared with the results of other comparison methods, the results of the proposed method could improve both the OA and Kappa values by considering the anisotropic relationship between classes. Especially, the proposed OMRF-AP model significantly improved segmentation accuracy on the HSR remote sensing images, where the average improvement was about 4%, and the maximum increment of OA and Kappa values could be about 10%.

In this paper, it was verified that APM was very useful for the MRF-based methods. In the OMRF-AP method, a heuristic setting approach was developed to set the APM, yet how to set APM with an unsupervised estimation according to the probability theory is still an important and open question, and we would explore this issue in the future works. Moreover, future experiments would also focus on finding clearer relationships between APM and geoscience knowledge.

**Supplementary Materials:** The following are available online at http://www.mdpi.com/2072-4292/11/23/2878/s1, Original Texture images of Figures 8 and 9; Original SPOT 5 image of Figure 10; Original aerial images of Figures 11 and 12; Original Gaofen-2 image of Figure 13; Original QuickBird image of Figure 15.

**Author Contributions:** All authors made significant contributions to this paper. Specifically, conceptualization, C.Z. and X.C.; methodology, C.Z., X.Y., X.X., and L.S.; software, X.P.; validation, X.X. and L.S.; formal analysis, C.Z. and X.C.; investigation, X.P.; resources, C.Z. and X.C.; data curation, C.Z. and X.P.; writing—original draft preparation, X.C. and C.Z.; writing—review and editing, all the authors; visualization, X.P. and X.Y.; Supervision, C.Z.

**Funding:** This research was funded by the National Natural Science Foundation of China under the Grant 41771375, 41571372, 41961053, and 31860182, the basic research funds for the Henan provincial universities, and the Fund of Reserve Talents for Young.

**Acknowledgments:** Tested aerial images are provided by associate Tiancan Mei of Wuhan University, China. Thank you very much.

**Conflicts of Interest:** The authors declare no conflict of interest.

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
