# Peer review of "An Object-Based Markov Random Field Model with Anisotropic Penalty for Semantic Segmentation of High Spatial Resolution Remote Sensing Imagery"

_remotesensing, doi:10.3390/rs11232878_

Round 1
Reviewer 1 Report
This is a novel technique and a good development of the MRF application.
Author Response
This is a novel technique and a good development of the MRF application.
We would like to thank the respected referee for the valuable comment. We have further improved the quality of this paper as best as we could.
Reviewer 2 Report
In this work an object-based Markov Random Field model (OMRF) extended with an anisotropic penalty matrix is proposed. The presented model, named OMRF-AP, is deployed for semantic segmentation of remote sensing high-resolution images. Although the proposed segmentation model is described in detail, the overall contribution of the current work is ambiguous, since the method requires a priori knowledge of the data and the experimental evaluation seems inadequate to prove its efficiency.
According to the authors, the confusion matrix is employed in order to calculate the values of the anisotropic penalty matrix (APM). However, it is unclear what is considered as ground truth in order to calculate the confusion matrix. In case that the initial pixel-based annotation is employed, then the application of the model is limited only to annotated data and cannot operate with unlabeled images or generalize to unknown data. In case that an estimation of the pixel-level ground truth is acquired via classical MRF, the performance of the proposed model is biased by the quality of the initial segmented result extracted with MRF. How is it guaranteed that MRF performs accurately for every examined case? Moreover, the number of depicted classes must be given as an input in the presented algorithm. This fact also confines the employment of OMRF-AP since it can not operate with previously unseen data. How would the model treat the case where a higher number of classes than those appeared in the image was given as input?
The overall evaluation of the proposed method is conducted in just four images. Only two of them are remote sensing data, while only one of them is considered as high-resolution image. The authors are suggested to examine the performance of their method thoroughly in a extended rich dataset or in remote sensing benchmarks. Moreover, statistics regarding the mean accuracy among the whole dataset would be helpful to evaluate the generalization ability of the model. Regarding the presented qualitative results, those are encouraging however the corresponding results presented in Table 4 imply that NED-MRF performs better than OMRF-AP for the cases of Fig. 8 and Fig. 9.
According to Table 5, the post-processing module (OMRF-APP) leads to minor improvements compared to OMRF-AP. Α comparison, in terms of computational time, between OMRF-AP and OMRF-APP would be useful in order to evaluate the efficiency of the post-process module. A more thorough evaluation should also include results from the application of the post-process step in the outcome of the rest comparative methods.
Overall, the contribution of this work is not well grounded by the experimental results. The out-performance of OMRF-AP compared to other MRF-based methods is inconsistent since NED-MRF presented higher performance in half of the examined cases. Furthermore, the advantages of the proposed method are unclear compared to other semantic segmentation methods, especially those including deep learning. OMRF-AP requires prior knowledge of the data, a lot of fine-tuning and provides coarse segmented results. On the contrary, deep learning techniques have proved their ability to provide fine segmentation results and generalize well in unseen data. Which are the benefits of the proposed method compared to deep learning models especially in the case of remote sensing data?
Author Response
Response to Reviewer 2 Comments
Point 1: In this work an object-based Markov Random Field model (OMRF) extended with an anisotropic penalty matrix is proposed. The presented model, named OMRF-AP, is deployed for semantic segmentation of remote sensing high-resolution images. Although the proposed segmentation model is described in detail, the overall contribution of the current work is ambiguous, since the method requires a priori knowledge of the data and the experimental evaluation seems inadequate to prove its efficiency.
Response 1: We would like to thank the respected referee for this valuable comment. In the revised version, we add a new section ‘Discussion’ to discuss the contributions and advantages of the proposed method (page 19). In this section, we also explain the priori information about the OMRF-AP and the difference between it and other semantic segmentation methods, such as deep learning. Moreover, two more experiments are also added to validate the superiority of the proposed method.
Point 2:According to the authors, the confusion matrix is employed in order to calculate the values of the anisotropic penalty matrix (APM). However, it is unclear what is considered as ground truth in order to calculate the confusion matrix. In case that the initial pixel-based annotation is employed, then the application of the model is limited only to annotated data and cannot operate with unlabeled images or generalize to unknown data. In case that an estimation of the pixel-level ground truth is acquired via classical MRF, the performance of the proposed model is biased by the quality of the initial segmented result extracted with MRF. How is it guaranteed that MRF performs accurately for every examined case? Moreover, the number of depicted classes must be given as an input in the presented algorithm. This fact also confines the employment of OMRF-AP since it can not operate with previously unseen data. How would the model treat the case where a higher number of classes than those appeared in the image was given as input?
Response 2: We would like to thank the respected referee for this valuable comment. In the revised version, we discuss more about the APM in the discussion section as well (page 19). Namely, When the OMRF-AP model is used for semantic segmentation, it only needs to set the number of land classes and the APM. Compared with the supervised method, such as the NED-MRF, the OMRF-AP does not need the training data with labeled classes and the ground truth. It can be used to the unlabeled images and unknow data when we set the number of classes and the APM. Specially, for the APM, although a heuristic setting approach is provided to set the APM, yet the suitable APMs are not unique. Hence, although the APM has many parameters, in addition to the heuristic setting approach, one can initialize the APM more flexible. Moreover, as the APM is used to reflect the relationship between certain land classes, the geoscience knowledge can also guide the setting of APM.
Point 3: The overall evaluation of the proposed method is conducted in just four images. Only two of them are remote sensing data, while only one of them is considered as high-resolution image. The authors are suggested to examine the performance of their method thoroughly in a extended
rich dataset or in remote sensing benchmarks. Moreover, statistics regarding the mean accuracy among the whole dataset would be helpful to evaluate the generalization ability of the model. Regarding the presented qualitative results, those are encouraging however the corresponding results presented in Table 4 imply that NED-MRF performs better than OMRF-AP for the cases of Fig. 8 and Fig. 9.
Response 3: We would like to thank the respected referee for this precise comment. In the revised version, two more remote sensing experiments are added to verity the effectiveness of the OMRF-AP in the experimental section. One is a very high spatial resolution remote sensing image, sized 2500×2500 with 0.1m spatial resolution (Page 15, figure 12). The other is a Gaofen-2 image that comes from a remote sensing dataset named Gaofen Image Dataset (GID), sized 1500×1500 with 3.2m spatial resolution (Page 16, figure 13). Moreover, we also have discussed why NED-MRF performs better than OMRF-AP for the cases of Fig. 8 and Fig. 9. in the discussion section (page 19).
Point 4: According to Table 5, the post-processing module (OMRF-APP) leads to minor improvements compared to OMRF-AP. Α comparison, in terms of computational time, between OMRF-AP and OMRF-APP would be useful in order to evaluate the efficiency of the post-process module. A more thorough evaluation should also include results from the application of the post-process step in the outcome of the rest comparative methods.
Response 4: We would like to thank the respected referee for this precise comment. In the revised version, the post-processing computational time is also considered in Table 6 (page 19). And the post-process step is also applied to the rest object-based comparative methods, the quantitative indicators are compared and demonstrated in Table 5 (page 17).
Point 5: Overall, the contribution of this work is not well grounded by the experimental results. The out-performance of OMRF-AP compared to other MRF-based methods is inconsistent since NED-MRF presented higher performance in half of the examined cases. Furthermore, the advantages of the proposed method are unclear compared to other semantic segmentation methods, especially those including deep learning. OMRF-AP requires prior knowledge of the data, a lot of fine-tuning and provides coarse segmented results. On the contrary, deep learning techniques have proved their ability to provide fine segmentation results and generalize well in unseen data. Which are the benefits of the proposed method compared to deep learning models especially in the case of remote sensing data?
Response 5: We would like to thank the respected referee for this valuable comment. In the revised version, more experiments of HSR remote sensing images are used to verify the superiority of the proposed method, and its advantages are further discussed in the new section ‘Discussion’ (page 19). In this section, we also discuss the benefits of the proposed method compared to deep learning in the case of remote sensing data. Namely, compared with other semantic segmentation methods, such as deep learning, the OMRF-AP has the complete theoretical foundation, and its APM has a clear and reasonable semantic interpretation. The proposed method is also very efficient as it does not need training processing.
Reviewer 3 Report
The paper presents an object-based Markov random filed model with anisotropic penalty for high resolution remote sensing image classification. The proposed method is interesting and performs well by distinguishing the differences in the interactions between any two land classes. Overall, this paper is well-presented and easy to follow, but there still have some issues need to be addressed. The justifications are detailed in the following:
(1) Section 1 should be improved. Authors are more focused on the problems of MRF itself, but the reason why MRF was chosen among many approaches to extract semantic information is not clearly discussed.
(2) Section 3 is not suitable to be a separate section. Since it introduces the parameter settings and algorithm of the proposed method, I would recommend moving section 3.1 to the experimental section, and integrating section 3.2 into section 2.2.
(3) The proposed method aims to improve the segmentation performance of HSR remote sensing images, but it is only evaluated on two HSR image patches, of which the SPOT5 experimental image size is only 438*438. More experiments on HSR remote sensing images are needed to prove the superiority of the proposed OMRF-AP.
(4) Please discuss why the proposed OMRF-AP achieves better experimental results on remote sensing images, but worse results on texture images, when compared with the NED-MRF.
(5) Conclusions should be extended. The section can be extended with statistical data obtained in experiment and with discussion about future experiments or improvements of method.
(6) Line 32: “Image” has incorrect font.
(7) Page 4: Formula numbers are not aligned.
(8) The format of the figure title is incorrect.
(9) Please check the format of the references.
Author Response
Response to Reviewer 3 Comments
The paper presents an object-based Markov random filed model with anisotropic penalty for high resolution remote sensing image classification. The proposed method is interesting and performs well by distinguishing the differences in the interactions between any two land classes. Overall, this paper is well-presented and easy to follow, but there still have some issues need to be addressed. The justifications are detailed in the following:
Point 1: Section 1 should be improved. Authors are more focused on the problems of MRF itself, but the reason why MRF was chosen among many approaches to extract semantic information is not clearly discussed.
Response 1: We would like to thank the respected referee for this valuable comment. In the revised version, the introduction is improved to discuss the reason why MRF was chosen among many approaches (page 2).
Point 2: Section 3 is not suitable to be a separate section. Since it introduces the parameter settings and algorithm of the proposed method, I would recommend moving section 3.1 to the experimental section, and integrating section 3.2 into section 2.2.
Response 2: We would like to thank the respected referee for this precise comment. In the revised version, we have moved section 3.1 to the experimental section, and integrated section 3.2 into section 2.2.
Point 3: The proposed method aims to improve the segmentation performance of HSR remote sensing images, but it is only evaluated on two HSR image patches, of which the SPOT5 experimental image size is only 438*438. More experiments on HSR remote sensing images are needed to prove the superiority of the proposed OMRF-AP.
Response 3: We would like to thank the respected referee for this valuable comment. In the revised version, two more remote sensing experiments are added to verity the effectiveness of the OMRF-AP in the experimental section. One is a very high spatial resolution remote sensing image, sized 2500×2500 with 0.1m spatial resolution (Page 15, figure 12). The other is a Gaofen-2 image that comes from a remote sensing dataset named Gaofen Image Dataset (GID), sized 1500×1500 with 3.2m spatial resolution (Page 16, figure 13).
Point 4: Please discuss why the proposed OMRF-AP achieves better experimental results on remote sensing images, but worse results on texture images, when compared with the NED-MRF.
Response 4: We would like to thank the respected referee for this precise comment. In the revised version, we have discussed the reason why NED-MRF performs slightly better than OMRF-AP on texture images in a new section ‘Discussion’ (page 19), and also discussed the advantages of the proposed method.
Point 5: Conclusions should be extended. The section can be extended with statistical data obtained in experiment and with discussion about future experiments or improvements of method.
Response 5: We would like to thank the respected referee for this valuable comment. In the revised version, we have extended the conclusion with statistical data obtained in experiment and with discussion about future experiments of the OMRF-AP (page 20). And the improvements of OMRF-AP are also discussed in the ‘Discussion’ section (page 19).
Point 6: Line 32: “Image” has incorrect font.
Response 6: We would like to thank the respected referee for this precise comment. In the revised version, we have corrected the font of this word.
Point 7: Page 4: Formula numbers are not aligned.
Response 7: We would like to thank the respected referee for this precise comment. In the revised version, formula numbers are aligned as best as we could (but improvements are quite limited).
Point 8: The format of the figure title is incorrect.
Response 8: We would like to thank the respected referee for this precise comment. In the revised version, we correct the format of the figure title according to the template of our journal.
Point 9: Please check the format of the references.
Response 9: We would like to thank the respected referee for this precise comment. In the revised version, we have checked the format of all the references.
Round 2
Reviewer 2 Report
This is a revised version of a previous work proposing a method named OMRF-AP for semantic segmentation of high-resolution remote sensing data. The presented method is based on an object-based Markov Random Field model (OMRF) combined with an anisotropic penalty matrix (APM). Although the authors extended further their work by adding two more experimental results and a more detailed discussion upon the benefits of the proposed model, yet the previously referred main ambiguities have not been sufficiently addressed.
According to the authors “the OMRF-AP does not need the training data with labeled classes and the ground truth”, while in another part of the text it is mentioned that OMRF-AP “only needs to set the number of land classes and the APM”. These two phrases are contradictory to each other and reflect the main problem in the framework of the proposed method. For the task of semantic segmentation, the number of classes is considered a priori knowledge (extracted from the ground truth) and thus OMRF-AP cannot generalize to previously unseen data, where the number of depicted classes is unknown. Even if a predefined fixed number of classes is assumed, there are tasks especially in case of remote sensing that semantic segmentation is exploited to detect environmental disasters, such as flood or pollution detection. A common case in this kind of problems is to cope with images where only one class is depicted (including background) and the model has to decide if it is a pollution incident or a look-alike. How would OMRF-AP handle that cases?
Moreover, as mentioned in the initial review comments, the fine-tuning of APM is based on the usage of the confusion matrix. This clearly implies that the proposed method utilizes the ground truth of the data and cannot generalize to unlabeled images. In the discussion part, the authors mention that the confusion matrix is utilized only in case of a heuristic setting approach, however all the presented results are based on this approach. As mentioned in the manuscript “one can initialize the APM more flexible”. How can one achieve a more flexible initialization that differs from the default one and does not follow the heuristic setting? Why did not the authors follow that approach? Results in Figure 14 are qualitative, a more thorough comparison should include the corresponding quantitative indicators. How did the authors conclude to the APM alternative notated as A’3?
Regarding the experimental evaluation, the method is tested in only six images. Why did the authors follow this approach instead of utilizing the entire GID dataset? The evaluation over a higher number of images would allow to extract more concrete results regarding the efficiency of the proposed model.
In conclusion, the overall contribution of this work remains ambiguous since the proposed method is based on a priori knowledge extracted from the data. The efficiency among with the generalization ability of the presented model remain also questionable as the experimental validation is limited to six images and utilizes the ground truth in each case. The advantages of OMRF-AP compared to deep learning techniques are also unclear. Indeed, deep learning approach requires a training process and a high number of data, however it has been proved that trained models can generalize well to unknown data in a large variety of semantic segmentation tasks. On the contrary, OMRF-AP lacks the ability to cope with unlabeled images.
Reviewer 3 Report
Thanks for your revision. I have no more questions to this manuscript for publication.
Author Response
Response to Reviewer 3 Comments
Point:Thanks for your revision. I have no more questions to this manuscript for publication.
Response: We would like to thank the respected referee for the valuable comment. We have further improved the quality of this paper as best as we could.